# How Do Language Models Speak Languages? A Case Study on Unintended Code-Switching

## Abstract

Unintended code-switching, which refers to the phenomenon where LLM unexpectedly switch languages, poses a fundamental challenge in the multilingual capabilities in LLMs. However, the fundamental properties of their underlying circuits, such as what they consist of, where they emerge in the network, and how to mitigate their effects, remain unexplored. Existing works on the mechanistic interpretability depend on additional training (e.g., sparse autoencoders) or manual annotation, both of which pose limitations in real-world scenarios. In this work, we introduce a scalable circuit discovery framework that causally localizes multilingual neurons, describes their functional patterns, and groups neurons into circuits. We find that the circuits for multilingual generation fall into two different regimes: a language regime which acts as a lingual key to detect language patterns, and a semantic regime which functions as a contextual value to retrieving language-agnostic semantics. These two regimes, in normal cases, converge smoothly to make final predictions, but in code-switching scenarios, semantics dominate the circuit, overriding typical language pathways and destabilizing outputs. Furthermore, we fine-tune the identified language sub-circuit ($\sim 0.019\%$ of all neurons), reducing the code-switching rate by $20.8\%$ with minimal parameter updates, validating the effectiveness of the discovered circuits for practical scalability. Our work serves as a preliminary exploration of multilingual generation circuits, offering actionable insights for neuron-based mechanistic interpretability.

## 1 Introduction

Large Language Models (LLMs) exhibit strong multilingual abilities in text understanding and generation (Alec Radford & Sutskever (2019), Hoffmann et al. (2022), Huang et al. (2023), Zhang et al. (2023), Zhao et al. (2024a)). Yet, recent studies reveal unintended code-switching[1]—mixing languages within a single utterance—during generation (DeepSeek-AI et al. (2025), Dubey et al. (2024), Lu et al. (2024)). For example: "Stephen Surjik est le réalisateur principal de ce film de 恐怖片" (English: Stephen Surjik is the lead director of this horror movie), a case generated by Qwen2.5-7B-Instruct model. Here, the French word peur ("horror") is incorrectly replaced by the Chinese 恐怖, yielding an unnatural switch.

While code-switching is natural in human multilingual communication (Auer & Wei (2007), Gumperz (1982)), model-generated switches often violate linguistic constraints and appear unpredictable. This raises a central mechanistic question: does code-switching emerge when internal reasoning—often in a dominant language such as English (Zhao et al. (2024b), Tang et al. (2024))—bypasses language-specific generation, leading to uncontrolled alternations? Despite its importance, the origins of code-switching remain largely unexplored. Prior work on multilingual mechanisms falls into two categories. Neuron-based methods identify language-activated neurons (Tang et al. (2024), Zhang et al. (2024b), Zhao et al. (2024b)) but lack causal evidence for their role. Feature-based methods train auxiliary modules such as sparse autoencoders (Marks et al. (2024), Lindsey et al. (2025)), but reconstructed features introduce interpretation gaps. Neither approach directly addresses the causal origins of code-switching or explains how multilingual decisions emerge in neural networks.

---

[1]Also known as *language confusion* (Marchisio et al. (2024))

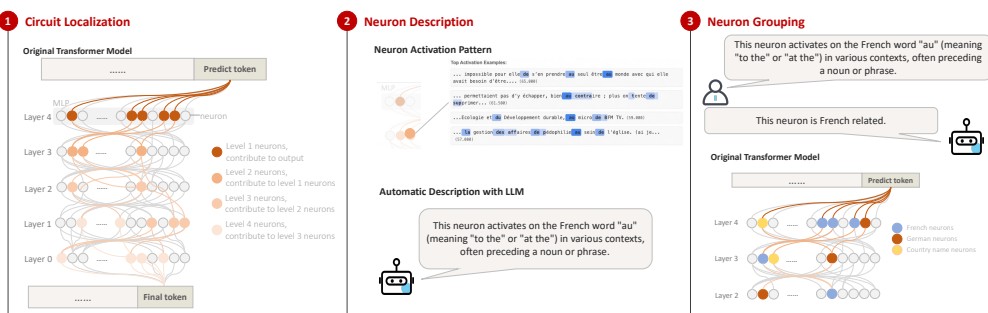

Figure 1: Overview of the proposed circuit discovery framework for interpreting unintended code-switching. Given an unmodified transformer model, we (1) discover circuits composed of hierarchical MLP neurons to explain their underlying mechanisms. (2) identify neurons' top activation samples, then employ an explainer model to generate textual descriptions of activation patterns, and (3) cluster functionally similar neurons. This structured grouping enhances circuit interpretability by consolidating semantically aligned components.

In this work, we address this gap with a scalable, causal circuit discovery framework that: (1) localizes multilingual neurons, (2) characterizes their functional patterns, and (3) groups them into circuits (Figure 1). Our method extends attribution patching (Nanda (2022)) from tracing component-to-output to tracing component-to-component attributions, yielding end-to-end causal circuits. Then, we employ LLMs to annotate neurons based on three criteria: a). the primary language of tokens that elicit the strongest activations, b). the presence of discernible semantic patterns in activation profiles, and c). whether the neuron exhibits selective promotion of specific token groups. Finally, LLMs are further used to cluster neurons with similar patterns into super-neurons, forming multilingual circuits.

The discovered circuits consist of two components: *language-specific super-neurons* that track contextual language, and *semantic super-neurons* that activate for language-agnostic concepts (e.g., horror-related terms). Figure 2 illustrates this with the prompt "il s'agit d'un survival horror avec un fort accent sur l'exploration et la", where the model completes with "peur" (horror). Our analysis highlights: (1) *Two-Step Generation*: the circuit first identifies the horror concept (semantic super-neuron) before engaging French-language neurons (Fig. 2a), showing a separation of conceptual and linguistic processing; (2) *Code-Switching Mechanism*: the semantic super-neuron directly drives the output (Fig. 2b), but manually up-weighting the language super-neuron (Fig. 2c) overrides this preference, proving these circuits are steerable and efficient. We provide examples of super-neurons within these sub-circuits in Figure 4.

We validate our framework through a series of experiments. Selective deactivation shows that disabling just 0.018% of language neurons leads to a 93.9% drop in multilingual generation, confirming their necessity. Fine-tuning only 0.19% of neurons reduces code-switching by 10%, demonstrating both efficiency and scalability. Attribution analysis further indicates that code-switching arises from imbalanced competition between semantic and language sub-circuits, and targeted suppression restores balance. We further conduct comprehensive analyses across languages and tasks, which confirm the robustness and generalizability of our findings.

To our knowledge, this is among the first works to provide causal evidence that code-switching arises from competition between semantic and language-specific circuits. Beyond advancing mechanistic understanding, our framework offers a diagnostic tool for linguistic errors and a pathway for targeted optimization in multilingual LLMs.

## 2 RELATED WORKS

Research on multilingualism in LLMs ranges from identifying language subspaces (Xie et al. (2024); Chang et al. (2022)) to locating discrete language-specific neurons and features (Zhang et al. (2024b); Zhao et al. (2024b); Tang et al. (2024); Lindsey et al. (2025); Ameisen et al. (2025)). These studies reveal where multilingual information may reside, but they generally lack causal validation of how such components drive cross-lingual behavior. Recent work on language confusion (Nie et al. (2025))

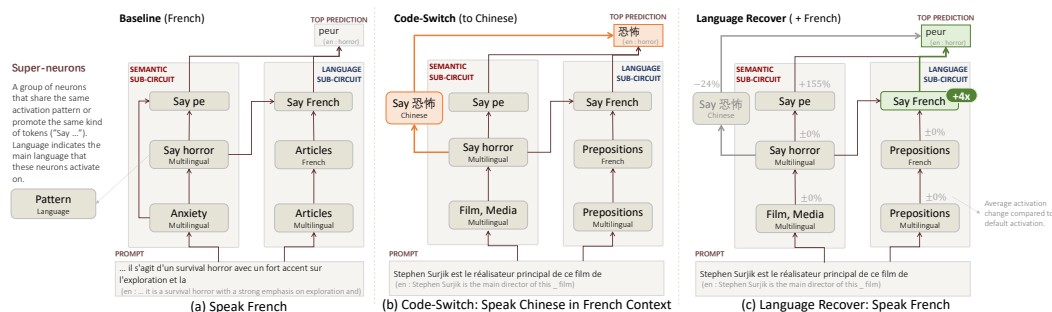

Figure 2: Circuits for: (a). speaking French normally; (b). code-switch (speak Chinese in French context); (c). recover French responses by activation manipulation. The light background boxes represent semantic (left) and language (right) sub-circuits. Dark inner boxes denote super-neurons, with bigger text indicating their activation patterns and smaller text specifying their dominant language. Normal French speaking circuit is made up of a language-agnostic semantic sub-circuit and a language-specific sub-circuit, whereas semantic information is passed down to language sub-circuit to make final prediction. Code-switching arises from erroneous direct connections between the semantic sub-circuit and output, bypassing language selection. Intervening on the language-indicating super-neuron suppresses code-switched tokens, restoring context-language consistency. The detailed circuits are Figure 7 and Figure 8 for normal French circuit and code-switching sample, respectively.

selects neurons by measuring log-probability shifts when injecting activations into the residual stream, showing that confusion-related neurons tend to appear in later layers. Our approach differs by using attribution-based neuron patching to recover circuits that causally influence both semantic and language-specific processing, highlighting competitive interactions rather than only confusion-related units.

More broadly, mechanistic interpretability has progressed from causal mediation analysis (Vig et al. (2020); Pearl (2001)) to circuit-level explanations via Path Patching (Olah et al. (2020); Wang et al. (2022); Hanna et al. (2023); Lieberum et al. (2023)), though these methods are costly at scale. Attribution Patching (Nanda (2022)) and its efficient variants (Kramár et al. (2024); Syed et al. (2023)) address this limitation and have been applied to multilingual settings (Dumas et al. (2025)). Whereas Dumas et al. (2025) argues for a sequential "language-then-concept" mechanism using activation patching, our neuron-level attribution approach instead reveals parallel pathways, where competition between semantic and language circuits explains unintended code-switching—offering a complementary view of multilingual processing.

## 3 PRELIMINARY

To analyze the intermediate mechanisms behind multilingual generation, we adopt attribution patching (Nanda (2022), Syed et al. (2023), Kramár et al. (2024)) to approximate causal effects with linear interventions. Conventional circuit analysis (Marks et al. (2024)) treats attribution patching as a single-step tool, yielding only localized causal evidence. We extend this approach through iterative causal tracing: (1) compute initial neuron-level attributions via gradient-based patching,[2] (2) recursively trace upstream inputs by patching connected neurons, and (3) terminate at embedding-layer representations, thereby reconstructing neuron-wise causal circuits for multilingual generation.

**Attribution Patching.** Given an LLM $M$, a contrastive pair of input $(x_{clean}, x_{corrupted})$, and metric $m$, let $n \in \mathbb{R}^d$ be a neuron (a column of the MLP down-projection) and $a \in \mathbb{R}$ its activation. Following Nanda (2022), the attribution of $n$ is:

$$\text{AttP}(m; a; x_{clean}, x_{corrupted}) = \nabla_a m|a = aclean(a_{clean} - a_{corrupted}) \quad (1)$$

Here $a_{clean}$ is $n$'s activation given input $x_{clean}$, and $\nabla_a m|_{a=a_{clean}}$ represents the gradient of $a$ when running on $x_{corrupted}$ but *intervening* by manually setting $a$ to $a_{clean}$. For example, given inputs $x_{clean}$="Angola is located in " and $x_{corrupted}$= "Angola liegt in " (French), we have metric

---

[2]Following Geva et al. (2021), "neuron" refers to a column of the MLP down-projection.

$$m(x) = \frac{LD_{patch} - LD_{corrupted}}{LD_{clean} - LD_{corrupted}} \tag{2}$$

where $LD = \text{Logit}[\texttt{Africa}] - \text{Logit}[\texttt{Afrika}]$ (Afrika is Africa in French), and $LD_{patch}$ refers to the logit difference when $a$ is patched to $a_{clean}$. Then a large value of $\text{AttP}(m; a; x_{clean}, x_{corrupted})$ indicates that the neuron is highly influential on the model's decision to output Africa rather than Afrika on this pair of inputs.

Attribution patching is efficient in identifying which neurons contribute to the final output since it requires only two forward passes and one backward pass under linearity assumptions, but it does not elucidate the underlying causes of these contributions—an aspect we argue is equally critical. To address this, we propose *hierarchical attribution patching*, which traces upstream neurons of $n$ by measuring the combined effect of paths traversing both the upstream neuron and $n$.

## 4 METHOD

In this section, we present a universal methodology for uncovering the mechanisms behind code-switching in multilingual LLMs. This section delves into three parts, including *Circuit Localization*, *Neuron Description* and *Neuron grouping*. Firstly, in Section 4.1, we outline the methodology to identify the neuron circuit within LLMs pertinent to different languages. Subsequently, in Section 4.2 and 4.3, we label and group these neurons in a fully automated manner, making our interpretation pipeline scalable and easy to reproduce. The full pipeline can be referred to in Figure 1.

### 4.1 CIRCUIT LOCALIZATION

**Hierarchical Attribution Patching.** The core intuition behind hierarchical attribution patching is to conceptualize a model as a computational graph, where each neuron functions as a component that reads from and writes to the residual stream (Nanda (2022); Geva et al. (2022)). Each late neuron aggregates inputs from earlier nodes, and the residual stream aggregates the outputs of all preceding components; to isolate one edge, we patch only the early neuron's output into the late neuron's input while freezing others.

Specifically, let $n_e$ denote the early neuron for a late neuron $n_l$, where the layer index $l_e$ for $n_e$ is shallower than $l_l$, their edge effect can be calculated as:

$$\text{AttP}_{\text{edge}}(m; a_e, a_l; x_{clean}, x_{corrupted}) = \nabla_{a_l} m|_{a_l = a_{l,clean}} (a_{e,clean} - a_{e,corrupted}) \tag{3}$$

where every value is projected onto the residual stream to maintain linearity and additivity.

We showcase the pseudo-code of our method in Algorithm 1, and a detailed mathematical derivation of Equation 1 and 3 in Appendix A.1.4. For the early node, we obtain its patched output in the residual stream by multiplying the scalar activation $n_e$ by the corresponding row of the MLP down-projection matrix. For the late node, we restrict computation to the edge mediated by the up-projection matrix rather than the gated projection, as the latter (e.g., via SiLU activation) would compromise linearity. To compute the gradient of the metric $m$ with respect to the residual stream—mediated solely by $n_l$, we first remove the SiLU output in $n_l$ and then multiply by the relevant row of the up-projection matrix, yielding $\nabla_{a_l} m|_{a_l = a_{l,clean}}$.

In practice, we identify "level-1" neurons directly influencing the output, then iteratively trace upstream neurons whose edge attribution exceeds $\epsilon$. This continues until either reaching the embedding layer or a maximum depth $L$ (set to 5 with $\epsilon = 0.001$), producing hierarchical circuits.

### 4.2 NEURON DESCRIPTION

Current neuron interpretation methods (Lee et al. (2024), Geva et al. (2022)) typically analyze neuron projection patterns, assuming each neuron promotes or suppresses token likelihoods via $P_{\text{vocab}}(n_{l,i}) = W_{\text{out}} \cdot W_{l,i}$, where $W_{\text{out}}$ is the output embedding matrix and $W_{l,i}$ the weights of neuron $n_{l,i}$. Yet neurons often display superposition, activating for multiple concepts across inputs, which static projections cannot fully capture. Complementary work (Choi et al. (2024)) shows that activation patterns can enrich projection-based analysis by automatically generating

meaningful neuron descriptions. Viewing feed-forward layers as key-value memories Geva et al. (2021), combining projections and activations offers a more complete view of neurons: keys correlate with training-text patterns, while values shape output distributions.

Accordingly, we rank tokens promoted by a neuron $n$ via dot products with unembedding matrix $e \cdot n$, then analyze activations across a multilingual corpus. Sentences are sorted by maximum token activation $T_{\max}(n_{l,i}) = t_j | \text{activation}(n_{l,i}, t_j) > \alpha$, where $\text{activation}(n_{l,i}, t_j)$ denotes the activation of neuron $n_{l,i}$ on the $t_j$-th token of the input, producing a top-$k$ set representative of $n$'s behavior. Integrating projection and activation patterns, we construct neuron profiles with LLMs: $P(n_{l,i}) = \text{LLM}(P_{\text{vocab}}(n_{l,i}), T_{\max}(n_{l,i}))$. This reveals the contexts that trigger $n$, the semantics it encodes, and the tokens it promotes. Implementation details, prompts, and validation experiments are provided in Appendices A.1.3, A.4, and A.3.3.

### 4.3 Neuron Grouping

To better interpret the circuit, we group neurons with related semantics and functions into super-neurons, following Lindsey et al. (2025). This yields a simplified view of the model's computation. Inspired by clustering in machine learning (Ester et al. (1998)), we pre-define categories such as language-specific (e.g., "German," "French") and context-specific (e.g., "Media," "Horror"). Each neuron's description from Section 4.2 is then classified by a judge model: $C(n_{l,i}) = LLM(P(n_{l,i}), c_1, c_2, \ldots, c_k)$, where $C(n_{l,i})$ is the assigned category and $c_1, \ldots, c_k$ are existing classes. If no match is found, a new category is generated and added to the taxonomy, producing a dictionary that maps labels to neuron groups. The classification prompt is in Appendix A.4.

### 5 Experiment

Our experiments are divided into five parts: circuit discovery and evaluation (Section 5.1), circuit validation (Section 5.2), discussion of code-switching through circuit competition (Section 5.3, fine-tuning neurons (Section 5.4), and an analysis of how input languages and tasks influence model's code-switching performance (Section 5.5). We conduct our experiments on Qwen2.5-7B-Instruct (Yang et al. (2024)) and LLaMA3.1-8B-Instruct (Dubey et al. (2024)) to validate the scalability of our method. We ran all experiments using four 80GB A100 GPUs. Additionally, the languages that we run our experiments on are French, Spanish, Russian, Chinese-simplified, Arabic, Japanese, Vietnamese, and Indonesian. The nuances of our language selection, hyper-parameter setting, dataset overview, and patching data construction are clarified in Appendix A.1.8, A.1.6, A.2.1, and A.2.2.

### 5.1 Circuit Discovery and Evaluation

**Circuit Discovery Details.** We analyze the last-token MLP neurons, which directly influence language selection (Geva et al. (2023), Zhao et al. (2024b)). Within these circuits, approximately 30% of neurons exhibit activation patterns across multiple languages, consistent with prior findings on multilingual core neurons (Zhang et al. (2024b)), while 60–70% remain monolingual, and this proportion is stable across both code-switched and non-code-switched samples. This suggests that code-switching is not driven by excessive multilingual interference but instead arises from other forms of internal competition (Appendix A.3.1).

**Circuit Evaluation.** We construct circuits for all eight languages and evaluate them using two metrics from Marks et al. (2024): faithfulness and completeness. Given a circuit $C$ and metric $m$, let $m(C)$ denote the average value of $m$ over a test dataset $D$ when running our model with all neurons outside of $C$ mean-ablated, i.e., set to their average value over data at each token from $D$. Let $m(\phi)$ denote the score when all neurons in the circuit are zero-ablated, and $m(M)$ the score of the full model with no intervention. We then measure faithfulness as $\frac{m(C)-m(\emptyset)}{m(M)-m(\emptyset)}$, which quantifies how much of the model's performance is captured by the circuit. Completeness is computed in the same way but using the circuit's complement $M \setminus C$, indicating how much behavior the circuit fails to account for.

Our results, shown in Figure 3, demonstrate two key findings: first, small circuits comprising fewer than 50 neurons explain the majority of the model's task performance (faithfulness), and second, ablating even a small subset of these critical neurons significantly degrades performance

(completeness). This sharp drop-off suggests these circuits operate as minimal functional units rather than redundant networks. Such sparsity confirms our method's ability to isolate interpretable, causally significant circuits that govern specific multilingual capabilities.

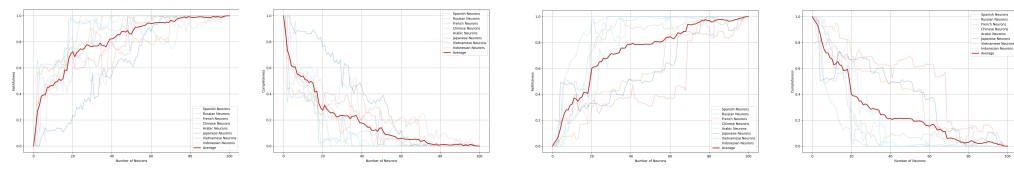

(a). Faithfulness and Completeness of Qwen2.5-7B-Instruct        (b). Faithfulness and Completeness of LLaMA3.1-8B-Instruct

Figure 3: Faithfulness and completeness of Qwen2.5-7B-Instruct (a) and LLaMA3.1-8B-Instruct (b) measured on $D$. Faint lines correspond to the different language circuits, with the average in bold and red. The ideal faithfulness for circuits is 1, while the ideal scores for their completeness is 0.

## 5.2 Language Neurons Validation for Different Circuits

We apply our method on Qwen2.5-7B-Instruct and LLaMA3.1-8B-Instruct, and obtain language circuits for the eight languages. To validate that these circuits are representative of the model's multilingual generation process, we performed intervention experiments on all the neurons in the circuit by inhibiting each of them (clamping them to a negative multiple of their original activation) on every token. See Appendix A.1.7 for discussion of the choice of intervention strengths and measuring the impact on the activations of neurons on the model output. Following Zhao et al. (2024b), we adopt the XLSum (Hasan et al. (2021)) dataset to evaluate multilingual performance as it requires the model to comprehend the input text and generate a coherent fragment. Specifically, we assess the performance of both models in corresponding languages when language-specific neurons in the circuits are deactivated versus when the same number of randomly sampled neurons is deactivated.

Table 1: Multilingual performance on XLSum when deactivating language-specific neurons in the circuit (Fr-neurons, Zh-neurons, Es-neurons, Ru-neurons, Ar-neurons, and Ja-neurons) and an equivalent number of randomly selected neurons (Random). We use $N_l$ to denote the neurons corresponding to the language $l$, whereas random neurons are represented as $N_{rand}$.

| Deactivated Neurons | Performances on Different Languages (%) | | | | | | | | | | |
|---|---|---|---|---|---|---|---|---|---|---|---|
| | Fr | Zh | Es | Ru | Ar | Ja | Vi | Id | $D_s$ | $D_{non-s}$ | $\Delta (\uparrow)$ |
| *Qwen2.5-7B-Instruct* | | | | | | | | | | | |
| None | 23.46 | 24.75 | 18.77 | 22.62 | 24.53 | 31.42 | 23.85 | 25.17 | - | - | - |
| $N_{rand}$ | 23.77 | 23.67 | 18.45 | 22.14 | 25.19 | 30.82 | 23.40 | 24.60 | - | - | - |
| $N_{Fr}$ | 6.53 | 23.66 | 17.58 | 18.40 | 23.16 | 27.90 | 17.72 | 22.09 | 16.93 | 1.08 | 15.85 |
| $N_{Zh}$ | 23.64 | 1.51 | 18.84 | 18.34 | 23.43 | 29.01 | 19.37 | 24.06 | 23.24 | 1.44 | 21.80 |
| $N_{Es}$ | 21.90 | 23.26 | 8.13 | 21.70 | 20.42 | 28.65 | 19.47 | 24.00 | 10.64 | 0.65 | 9.99 |
| $N_{Ru}$ | 22.64 | 23.27 | 18.47 | 3.87 | 23.59 | 30.22 | 20.37 | 19.02 | 18.75 | 0.95 | 17.80 |
| $N_{Ar}$ | 23.57 | 23.76 | 18.56 | 22.47 | 8.20 | 30.21 | 23.42 | 24.60 | 16.33 | 1.85 | 15.84 |
| $N_{Ja}$ | 22.13 | 22.18 | 16.81 | 21.04 | 22.70 | 2.58 | 21.68 | 24.35 | 28.84 | 1.01 | 27.83 |
| $N_{Vi}$ | 22.89 | 24.06 | 18.15 | 22.41 | 23.15 | 30.45 | 3.48 | 24.33 | 20.37 | 0.24 | 20.13 |
| $N_{Id}$ | 23.52 | 23.61 | 18.61 | 22.24 | 24.23 | 31.15 | 23.82 | 8.99 | 16.18 | 2.25 | 13.93 |
| *LLaMA3.1-8B-Instruct* | | | | | | | | | | | |
| None | 24.24 | 29.70 | 20.97 | 25.15 | 25.46 | 33.11 | 20.86 | 25.61 | - | - | - |
| $N_{rand}$ | 24.89 | 29.82 | 20.73 | 24.10 | 26.68 | 33.55 | 20.89 | 25.71 | - | - | - |
| $N_{Fr}$ | 0.66 | 29.56 | 19.87 | 25.05 | 25.12 | 32.92 | 17.95 | 21.56 | 23.58 | 1.07 | 22.51 |
| $N_{Zh}$ | 22.39 | 1.50 | 19.53 | 24.45 | 24.38 | 32.57 | 22.60 | 24.18 | 28.20 | 0.33 | 27.87 |
| $N_{Es}$ | 23.21 | 23.55 | 2.73 | 23.07 | 24.38 | 31.69 | 19.29 | 21.55 | 18.24 | 1.24 | 17.00 |
| $N_{Ru}$ | 23.71 | 29.12 | 19.80 | 6.20 | 24.78 | 33.00 | 20.56 | 25.75 | 18.95 | 1.31 | 17.64 |
| $N_{Ar}$ | 22.78 | 26.81 | 19.37 | 24.94 | 2.05 | 32.89 | 21.28 | 27.08 | 23.41 | 0.06 | 23.35 |
| $N_{Ja}$ | 23.67 | 28.36 | 19.77 | 24.83 | 23.50 | 4.34 | 23.49 | 26.69 | 28.77 | -1.00 | 27.77 |
| $N_{Vi}$ | 23.47 | 28.18 | 21.53 | 23.69 | 24.66 | 30.58 | 7.41 | 24.14 | 13.45 | 0.98 | 12.47 |
| $N_{Id}$ | 24.01 | 29.26 | 20.18 | 23.93 | 26.38 | 32.99 | 21.25 | 5.13 | 20.48 | 1.25 | 19.23 |

Table 1 shows how deactivating language-specific neurons selectively impairs performance in their corresponding languages, where $D_s$ represents performance declination of the corresponding language (e.g., the declination of French performance when deactivating French neurons) and $D_{\text{non-s}}$ represents the average performance declination of the other three languages. We quantify language specificity through the metric $\Delta = D_s - D_{\text{non-s}}$, and higher $\Delta$ values directly reflect stronger, more specialized linguistic processing in identified neurons.

Notably, deactivating just $0.018\%$ of these neurons produces sharp, selective performance drops in the corresponding language (e.g., French declines by up to $97.3\%$ in LLaMA, Chinese by $93.9\%$ in Qwen), while other languages remain largely unaffected and random neuron ablations cause minimal loss. Interestingly, the magnitude of loss varies across languages and models: Qwen shows stronger declines in typologically distant or lower-resource languages, while LLaMA exhibits more balanced but still severe drops, suggesting differences in training data coverage and model inductive biases.

Together, these results reveal that language circuits are not only necessary for multilingual generation but also shaped by linguistic diversity and corpus distribution. Since neurons operate independently,different languages should also function independently, suggesting that **language confusion is not due to competition between languages**, but rather arises from **the competition between language-specific circuits and semantic circuits**. This highlights the need for deeper exploration of the underlying mechanisms behind code-switching.

## 5.3 DISCUSSION OF CODE-SWITCHING THROUGH CIRCUIT COMPETITION

Attribution quantifies the contribution of components by measuring the change in output loss or logits when their clean activations are restored Ameisen et al. (2025); Nanda (2022). Using this lens, we find that code-switching arises from internal competition between language and semantic circuits, where reduced dominance of language attribution allows semantic signals to intervene. To test this, we sample 10 code-switched and 10 non-code-switched examples per language, build three-level circuits, and label super-neurons as semantic or language based on their group profiles. Averaged attribution ratios (Table 2), computed by dividing the mean attribution of language neurons by that of semantic neurons, support this view: language circuits clearly dominate in non-switch cases ($2.37\times$), but their lead narrows in switch cases ($1.21\times$). The ratio never falls below $1.0$, suggesting these are "potential" switch points where the code-switched token is probable but not yet top-ranked. Together, these results show that code-switching reflects a weakened dominance of language circuits and increased competition from semantic circuits. A mathematical derivation of this competence is provided in Appendix A.1.5.

To directly validate the causal role of semantic circuits, we suppress their activations at the token preceding the confusion point (i.e., the token position where code-switching happens) in code-switched samples. We then compare prediction probabilities for the original code-switched token (CSW token) versus a non-confusion token aligned with the intended language (generated with temperature 0.0 and aligned with the intended language). As shown in Table 3, suppression reduces confusion-token probabilities by $99.67\%$ ($0.06 \rightarrow 0.0002$), while boosting non-confusion token probabilities by $67.68\%$ ($0.3964 \rightarrow 0.6647$). These interventions provide strong causal evidence that semantic circuits drive code-switching by competing with language circuits.

We also provide another perspective with LogitLens (nostalgebraist (2020)) in Appendix A.3.2 to give additional insights on how layer-wise representations reflect the conversion from monolingual outputs to multilingual outputs in deeper layers, and how code-switched samples show different conversion pattern compared with non-code-switched samples. These results are contributory in understanding the nuances and mechanisms behind model's code-switching phenomenon.

## 5.4 CODE-SWITCH EVALUATION AND REDUCTION

Building on the mechanism uncovered in Section 5.3, we now turn to evaluating how frequently code-switching occurs in practice and how precisely fine-tuning identified neurons can mitigate it. We evaluate unintended code-switching using the Language Confusion Benchmark (LCB) (Nie et al. (2025)), which builds on monolingual prompt setups from Tramm et al. (2024) and spans a wide range of languages. Performance is measured by Line-level Pass Rate (LPR), the percentage of responses where every line remains in the target language. Following the original setup, we use temperature =

| Samples | Ratio (Language / Semantic) |
|---------|------------------------------|
| NCSW | 2.3717 |
| CSW | 1.2073 |

Table 2: Language vs. Semantic attribution ratio in non-code-switched circuits and code-switched circuits.

| Type | $\bar{P}$ before | $\bar{P}$ after |
|------|-----------|----------|
| CSW token | 0.0608 | 0.0002 |
| NCSW token | 0.3964 | 0.6647 |

Table 3: Model's prediction probabilities for the original code-switched token and a non-code-switched token (generated with temperature 0.0 and aligned with the intended language), before and after intervention.

0.3, top_p = 0.75, regenerate each response 10 times, and extend the maximum output length to 1024 tokens to capture longer generations where code-switching is more likely. We provide additional experiments in Appendix A.3.6 to discuss how temperature influences model code-switching.

Experiments on Qwen2.5-7B-Instruct and LLaMA3.1-8B-Instruct (Table 4, first column under each model's name) reveal consistently high code-switching rates across languages, suggesting the phenomenon is not tied to specific language pairs but reflects fundamental architectural properties of multilingual LLMs, requiring systematic rather than language-specific solutions.

Table 4: LPR for the original model ("Baseline"), fine-tuning random neurons ("$N_{Rand}$"), and fine-tuning language-specific neurons ("$N_{Lang-Spec}$").

| Trained Neurons | Line-level Pass Rate (↑) | | | | | | | | |
|-----------------|--------|--------|--------|--------|--------|--------|--------|--------|---------|
| | **Ar** | **Es** | **Fr** | **Ja** | **Ru** | **Zh** | **Vi** | **Id** | **Average** |
| *Qwen2.5-7B-Instruct* | | | | | | | | | |
| Baseline | 96.41% | 96.67% | 95.18% | 88.78% | 98.08% | 92.10% | 97.38% | 82.96% | 93.45% |
| $N_{Rand}$ | 91.87% | 96.13% | 94.25% | 87.70% | 95.66% | 93.65% | 98.20% | 86.70% | 93.02% |
| $N_{Lang-Spec}$ | **98.50%** | **97.56%** | **96.62%** | **98.49%** | **99.16%** | **96.35%** | **99.10%** | **92.60%** | **97.30%** |
| *LLaMA3.1-8B-Instruct* | | | | | | | | | |
| Baseline | 99.25% | 97.63% | 97.86% | 96.00% | 98.40% | 94.55% | 99.70% | 83.50% | 95.99% |
| $N_{Rand}$ | 99.37% | 96.63% | 97.97% | 95.70% | 99.40% | 93.30% | 99.80% | 84.20% | 95.83% |
| $N_{Lang-Spec}$ | **99.87%** | **98.50%** | **98.63%** | **97.80%** | **99.87%** | **96.55%** | **99.90%** | **88.20%** | **97.36%** |

Building on this quantitative understanding, we next explore whether fine-tuning targeted neurons can mitigate code-switching. Because language-specific neurons constitute only 0.19% of the model, tuning them incurs minimal computational cost while preserving overall task performance. We have detailed the process of training data construction in Appendix A.2. For comparison, we also perform full-parameter fine-tuning on the same training data as neuron tuning, but the results were less effective; details are provided in the Appendix A.3.5.

The results are shown in Table 4 (detailed in Appendix A.3.4). While fine-tuning random neurons has little impact, our approach substantially improves LPR for both models across languages by updating only a few hundred neurons, highlighting the precision of the identified language-specific neurons. We also benchmark our method against standard baselines, including few-shot prompting and full-parameter fine-tuning, with results provided in Appendix A.3.7.

## 5.5 IMPACT OF INPUT TYPES AND TASKS ON CODE-SWITCHING PERFORMANCE

Building on the findings from Section 5.4, we now analyze how different input types affect model code-switching. Our experiments focus primarily on Arabic and Japanese, which tend to exhibit more frequent code-switching due to being low-resource languages. Arabic, with its rich history of linguistic exchange (Hamed et al. (2025)), and Japanese, being linguistically distant from the high-resource languages, serve as representative low-resource languages for this study. We first evaluate Qwen and LLaMA's baseline code-switching performance on Arabic and Japanese tasks, including **translation** (Costa-jussà et al. (2022)), **instruction following** (Zhang et al. (2024a)), **QA** (Longpre et al. (2021), So et al. (2022), Artetxe et al. (2020)), and **conversation** (Ding et al. (2023)). The details of the datasets we use are clarified in Appendix A.1.9.

**Evaluation Method**: We evaluate using ***Language Consistency Rate*** (LCR) — the proportion of model outputs that remain entirely in the intended language without unintended code-switching. (details in Appendix A.1.9). We employ generation parameters (top_p=0.8, top_k=20, temperature=0.7, presence_penalty=1.5) to balance output diversity with a minimal code-switching rate and generate 128 responses for each question to compute the final LCR.

Table 5: Model's LCR when fine-tuning different types of neurons. A higher LCR indicates a lower unintended code-switching occurrence.

| Model | Language | Methods | LCR on Different Tasks (↑) | | | | |
|---|---|---|---|---|---|---|---|
| | | | Translation | IF | QA | Conversation | **Average** |
| Qwen2.5-7B-Instruct | Ar | None (Baseline) | 47.93% | 65.00% | 54.19% | 54.37% | 53.94% |
| | | $N_{Ar}$ | **67.23%** | **77.37%** | **81.86%** | **74.46%** | **75.23%** |
| | Ja | None (Baseline) | 74.95% | 49.31% | 50.53% | 32.48% | 51.82% |
| | | $N_{Ja}$ | **91.69%** | **75.55%** | **99.52%** | **62.11%** | **82.22%** |
| LLaMA3.1-8B-Instruct | Ar | None (Baseline) | 88.57% | 74.75% | 92.67% | 59.99% | 79.00% |
| | | $N_{Ar}$ | **95.97%** | **97.18%** | **95.04%** | **96.06%** | **96.06%** |
| | Ja | None (Baseline) | 89.49% | 61.43% | 79.9% | 46.95% | 69.44% |
| | | $N_{Ja}$ | **97.25%** | **89.05%** | **99.68%** | **87.71%** | **93.42%** |

**Results**: Table 5 demonstrates that targeted fine-tuning of just 50 Arabic-specific neurons increases overall LCR by 21.29%, confirming both the effectiveness and specificity of our identified circuit.

From the results, we can conclude some observations:

- **Language Differences**: Despite being a low-resource language, Arabic generally causes lower code-switching (higher LCR) than Japanese. However, LCR varies significantly across tasks for both languages.
- **Task Differences**: Task type strongly affects LCR. Translation and QA typically yield higher LCR, likely because they produce shorter, finite responses with well-defined outputs. In contrast, Instruction Following and Conversation are open-ended and often result in longer outputs—conditions under which models are more prone to code-switching.

These findings suggest that both the language and the structure of the task significantly influence a model's tendency to code-switch. In particular, longer and less constrained outputs increase the risk of unintended code-switching. Nonetheless, the results across diverse tasks further validate the generalizability of our neuron identification and precise fine-tuning results.

## 6 LIMITATION

Understanding the sophisticated mechanisms behind LLMs' multilingualism remains an ongoing challenge. While we validated our circuits through deactivation experiments and fine-tuning, further work is needed to extend the discovery pipeline to support cross-token and cross-structure (e.g., attention block) attribution patching. Another promising direction is to investigate how models integrate reasoning with multilingual knowledge, which could enhance performance in tasks such as cross-lingual transfer. We hope these insights provide a valuable foundation for future research.

## 7 CONCLUSION

In this work, we examine the intricate mechanisms of LLM multilingual generation, using code-switching as a special case. We validate our circuit discovery framework through deactivation methods, showing that LLMs generate multilingual responses via two paths: a semantic path (primarily in English) and a language-specific path that identifies linguistic patterns and promotes the context language neurons for final predictions. Our systematic evaluation quantifies code-switching rates, localizes its causes, and demonstrates that fine-tuning language-specific neurons reduces unintended code-switching by 20.8%, without compromising performance. By bridging circuit analysis with multilingual behavior, our framework provides new capabilities for diagnosing and mitigating language-specific generation errors in LLMs.

ETHICS STATEMENT

In this paper, we used large language models (LLMs) solely to assist with language polishing. LLMs did not contribute to the research motivations, framework design, or method implementation. We affirm that our code and methodology do not intentionally introduce discrimination, bias, unfairness, or risks related to misuse, privacy, security, legal compliance, or research integrity. Nonetheless, we acknowledge that existing language datasets and pretrained models may embed inherent biases, which could be inherited by the multilingual models employed in our work.

REPRODUCIBILITY STATEMENT

To support reproducibility, we provide comprehensive details regarding our proposed framework in both the main text and the appendix.

**Methodology**. A thorough description of our method is presented in Section 4, including the hierarchical attribution patching (Section 4.1), neuron description (Section 4.2), neuron grouping (Section 4.3), and training details (Appendix A.1.6, Appendix A.2.1). We provide figures (Figure 1, Figure 2) and pseudo-code (Algorithm 1) to enhance the readability.

**Datasets**. We employ open-source datasets for circuit discovery and training (Appendix A.2.1) and utilize open-source benchmarks for evaluation (Appendix A.2.1, Table 6). All datasets and benchmarks referenced in the paper are publicly accessible and can be downloaded from the Hugging Face platform [3].

**Experiment Details**. Detailed information on method implementation is provided in Appendix A.1. This includes computation details, hyper-parameter configurations, choices of intervention strengths, and language selection. The dataset overview (Appendix A.2.1), LLM evaluation prompts (Appendix A.4), and ablation experiment details (Appendix A.3) used in the experiments are included in the appendix for ease of reproduction.

Additionally, we provide the training and evaluation code in the supplementary materials. Should the paper be accepted, we commit to making the full source code for our approach publicly available.

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

# A APPENDIX

## A.1 IMPLEMENTATION DETAILS

### A.1.1 EXAMPLES OF SUPER-NEURONS

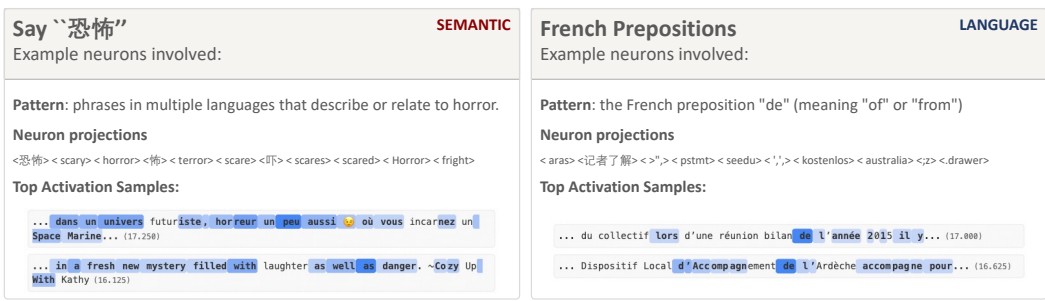

Figure 4: Example super-neurons and neurons which participate in the code-switching circuit. Neurons are active on tokens shaded in blue, where darker color indicates stronger activation value.

### A.1.2 ALGORITHM

Below we show the pseudo-code of our Hierarchical Attribution Patching Circuit Discovery method.

---

**Algorithm 1** Hierarchical Attribution Patching Circuit Discovery

---

**Require:** Model $M$, clean/corrupted data $(X_r, A_r)/(X_c, A_c)$, threshold $\epsilon$, max level $L$, Attribution Patching function $f : \mathcal{N} \times \mathcal{N} \to \mathbb{R}^{l \times d}$, where $\mathcal{N}$ is the set of neurons, $f(s, e)$ computes *direct attribution* (if $e =$ output) or *edge effect* (if $e \in \mathcal{N}$)

**Ensure:** Hierarchical circuit neurons $\{C_1, ..., C_k\}$ where $k \leq L$

1: **function** MAIN($M, X_r, X_c, A_r, A_c, \epsilon, L$)
2:     $C_1 \leftarrow \{(l, n) \mid f(n, \text{output}) > \epsilon\}$              ▷ Level 1: Output-attributing neurons
3:     **for** $level \leftarrow 2$ to $L$ **do**
4:         $C_{level} \leftarrow \emptyset$
5:         **for** $(l_{end}, n_{end}) \in C_{level-1}$ **do**
6:             **for** $(l_{src}, n_{src})$ where $l_{src} < l_{end}$ **do**
7:                 **if** $f((l_{src}, n_{src}), (l_{end}, n_{end})) > \epsilon$ **then**
8:                     $C_{level} \leftarrow C_{level} \cup \{(l_{src}, n_{src})\}$
9:                 **end if**
10:             **end for**
11:         **end for**
12:         **if** $C_{level} = \emptyset$ **then break**
13:         **end if**
14:     **end for**
15:     **return** $\{C_1, ..., C_{level}\}$
16: **end function**

---

### A.1.3 COMPUTATION DETAILS

In this section we provide details from Geva et al. (2022) that demonstrate that MLP neurons promote or suppress the likelihood of tokens.

We start from Equation 4:

$$\text{MLP}^l(\mathbf{x}^l) = \sum_{i=1}^{d_{\text{MLP}}} \sigma(x^l \cdot \mathbf{k}_i^l)\mathbf{v}_i^l = \sum_{i=1}^{d_{\text{MLP}}} m_i^l \mathbf{v}_i^l \qquad (4)$$

Thus, we can consider the update form $\mathrm{MLP}^l$ as $d_{\mathrm{MLP}}$ *sub-updates*, each sub-update being $m_i^l \mathbf{v}_i^l$.

We can then analyze the influence that each sub-update has on the output distribution, or the probability of generating token $\omega \in V$ (taken from Geva et al. (2022)):

$$p(\omega | \mathbf{x}^l + m_i^l v_i^l, E) = \frac{\exp(\mathbf{e}_\omega \cdot \mathbf{x}^l + \mathbf{e}_\omega \cdot m_i^l \mathbf{v}_i^l)}{Z(E(\mathbf{x}^l + m_i^l \mathbf{v}_i^l))} \propto \exp(\mathbf{e}_\omega \cdot \mathbf{x}^l) \cdot \exp(\mathbf{e}_\omega \cdot m_i^l \mathbf{v}_i^l) \tag{5}$$

where $\mathbf{e}_\omega$ is the token embedding of $\omega$, and $Z$ is the softmax normalization factor. This indicates that when $\mathbf{e}_\omega \cdot m_i^l \mathbf{v}_i^l > 0$, the likelihood of $\omega$ increases, while $\mathbf{e}_\omega \cdot m_i^l \mathbf{v}_i^l < 0$ decreases the likelihood.

### A.1.4 DERIVATIONS OF EQUATION 1 AND 3

Let $i$ represent the index of a Chinese word in the vocabulary, and $j$ the index of the corresponding French word (e.g., at position $i$, we have "非洲" and at position $j$, "Afrika").

Define $e_i$ as the one-hot vector in $\mathbb{R}^V$ where the $i$-th position is 1 and the rest are 0. The logits corresponding to the Chinese and French tokens can then be written as:

$$\mathrm{Logits}[非洲] = \mathrm{Logits}^T \cdot e_i, \quad \mathrm{Logits}[\mathrm{Afrika}] = \mathrm{Logits}^T \cdot e_j \tag{6}$$

Given a neuron with activation $a$, let $a_0 = a_{\mathrm{clean}}$ represent the activation for a Chinese input, and $a_1 = a_{\mathrm{corrupt}}$ represent the activation for a French input. Let $x_0$ denote the first $n-1$ tokens of the Chinese input and $x_1$ denote the first $n-1$ tokens of the French input.

$\mathrm{Logits}(a, x)$ represents the logits produced by the model when the input is $x$ and the targeted neuron activation is forced to be $a$. So: - $\mathrm{Logits}_{\mathrm{clean}} = \mathrm{Logits}(a_0, x_0)$ represents the model's original prediction on the clean Chinese input, using the neuron activation naturally induced by the Chinese context. - $\mathrm{Logits}_{\mathrm{corrupt}} = \mathrm{Logits}(a_1, x_1)$ represents the model's prediction on the fully corrupted French input, using the neuron activation naturally induced by the French context. - $\mathrm{Logits}_{\mathrm{patch}} = \mathrm{Logits}(a_0, x_1)$ represents a causal intervention where the French input is used but the neuron activation is overwritten with the activation from the Chinese instance, thereby isolating the causal effect of that neuron on the final logits.

Thus, the logit differences (LD) are:

$$\begin{aligned} LD_{\mathrm{patch}} &= \mathrm{Logits}_{\mathrm{patch}}[非洲] - \mathrm{Logits}_{\mathrm{patch}}[\mathrm{Afrika}] \\ &= \mathrm{Logits}_{\mathrm{patch}}^T \cdot (e_i - e_j) = \mathrm{Logits}(a_0, x_1)^T \cdot (e_i - e_j) \\ LD_{\mathrm{corrupt}} &= \mathrm{Logits}(a_1, x_1)^T \cdot (e_i - e_j) \\ LD_{\mathrm{clean}} &= \mathrm{Logits}(a_0, x_0)^T \cdot (e_i - e_j) \end{aligned} \tag{7}$$

Given the formula above, the metric $m$ in Equation (2) can be written as:

$$m = \frac{(\mathrm{Logits}(a_0, x_1) - \mathrm{Logits}(a_1, x_1))^T \cdot (e_i - e_j)}{(\mathrm{Logits}(a_0, x_0) - \mathrm{Logits}(a_1, x_1))^T \cdot (e_i - e_j)} \tag{8}$$

**Deriving Equation (1)** : Think of $m$ as a function of the patched activation $m = f(a)$. Holding all other activations fixed, we apply a first-order Taylor expansion around the corrupted activation $a$, so that

$$f(a) \approx f(a_1) + \nabla_a f(a_1)^\top \cdot (a - a_1) \tag{9}$$

Setting $a = a_0$ gives

$$\Delta m = f(a) - f(a_1) = f(a_1) + \nabla_a f(a_1) \cdot (a0 - a_1) - f(a_1) = \nabla_a f(a_1)^\top \cdot (a_0 - a_1) \tag{10}$$

This yields Equation (1):

$$AttP(m; a; x_0, x_1) = \nabla_a m|_{a=a_0}(a_0 - a_1) \tag{11}$$

This quantity measures the contribution of neuron activation $a$ to the final logits, i.e., the final node of the computation graph.

**Deriving Equation (3)**: Let $n_{l,i}$ be a neuron in layer $l$, indexed at $i$-th row in this layer's MLP down projection matrix. We can then denote early neuron and late neuron as $n_{l_1,i}$ and $n_{l_2,j}$ where $l_1 < l_2$.

Because of the linearity we assumed through out the paper, we can consider the sum of the path attribution patch values over all start nodes (including the embedding) should equal the end node's total attribution patch value.

Thus:

$$a_{l_2,j} = a_{l_1,i} + \sum (\text{other residual terms}) \tag{12}$$

Although the activation values $a_{l_1,i}$ and $a_{l_2,j}$ typically pass through nonlinear activation functions (such as ReLU, GELU, etc.), the residual connection itself is based on a linear addition of information. This means that, even with nonlinear activation functions, the residual connection simply adds the output from earlier layers to the subsequent layers. Therefore, during backpropagation, the gradient flow through this linear addition is independent of the nonlinear activation functions. Specifically, the contribution of the early neuron to the late neuron is linear, as the gradient with respect to the early neuron's output is constant. This leads to the following relationship

$$\frac{\partial a_{l_2,j}}{\partial a_{l_1,i}} = I \tag{13}$$

This relationship holds for each neuron at the same level because the residual connection ensures that the influence of the early neuron on the late neuron remains linear, unaffected by the nonlinearities in intermediate layers.

Given the conclusion above, we can continue calculating the attribution of $n_{l_1,i}$ on $m$ through $n_{l_2,j}$. Since $m = f(a_{l_2,j}, \text{other residual outputs})$, and $a_{l_2,j}$ depends on $a_{l_1,i}$, we obtain

$$\frac{\partial m}{\partial a_{l_1,i}} = \frac{\partial m}{\partial a_{l_2,j}} \frac{\partial a_{l_2,j}}{\partial a_{l_1,i}} + \sum_{\text{other paths p}} \frac{\partial m}{\partial a_p} \frac{\partial a_p}{\partial a_{l_1,i}} \tag{14}$$

To isolate the attribution only along the path $n_{l_1,i} \to n_{l_2,j} \to m$, we retain the mediated term:

$$\frac{\partial m}{\partial a_{l_1,i}} = \frac{\partial m}{\partial a_{l_2,j}} \frac{\partial a_{l_2,j}}{\partial a_{l_1,i}} \tag{15}$$

Using $\frac{\partial a_{l_2,j}}{\partial a_{l_1,i}} = I$:

$$AttP_{n_{l_1,i} \to n_{l_2,j} \to m}(m; a_{l_1,i}, a_{l_2,j}; x_0, x_1) = \nabla_{a_{l_2,i}} m|_{a_{l_2,i} = a_{l_2,i,x_1}} (a_{l_1,i,x_0} - a_{l_1,i,x_1}) \tag{16}$$

which corresponds to the Equation (3):

$$AttP_{\text{edge}}(m; a_{l_1,i}, a_{l_2,j}; x_0, x_1) = \nabla_{a_{l_2,i}} m|_{a_{l_2,i} = a_{l_2,i,x_1}} (a_{l_1,i,x_0} - a_{l_1,i,x_1}) \tag{17}$$

### A.1.5 DERIVATION OF COMPETENCE

Given Equation 8, we can treat $LD_{\text{corrupt}}$ and $LD_{\text{clean}}$ as constants when differentiating, hence we obtain:

$$\frac{\partial m}{\partial a}\bigg|_{a=a_0, x=x_1} = \frac{1}{(\text{Logits}(a_0, x_0) - \text{Logits}(a_1, x_1))^T \cdot (e_i - e_j)} \left( \frac{\partial \text{Logits}(a, x)}{\partial a}\bigg|_{a=a_0, x=x_1} \right)^T \cdot (e_i - e_j) \tag{18}$$

Hence, the attribution Attr is:

$$\text{Attr} = \frac{a_0 - a_1}{(\text{Logits}(a_0, x_0) - \text{Logits}(a_1, x_1))^T \cdot (e_i - e_j)} \left( \frac{\partial \text{Logits}(a, x)}{\partial a}\bigg|_{a=a_0, x=x_1} \right)^T \cdot (e_i - e_j) \tag{19}$$

The activation $a$ represents the neural response to a specific input, which influences the output logits. Specifically, $\frac{\partial m}{\partial a}$ represents how much a change in the activation $a$ affects the difference between the logits for the Chinese and French tokens. This term measures the contribution of a particular neuron

to the output, capturing the extent to which the neuron's activation influences the logit difference, or the "decision" for the model's output.

Now, let's introduce two types of neurons: one representing language-specific patterns whose activations are denoted by $a_l$, and the other representing semantic patterns whose activations are denoted by $a_s$. Notably, the denominator $(\text{Logits}(a_0, x_0) - \text{Logits}(a_1, x_1))^T \cdot (e_i - e_j)$ is independent of which neuron we choose for patching (because it doesn't require patching and is calculated over fixed input), so it cancels out when comparing the ratio.

Thus, the ratio of attributions for the language and semantic neurons is:

$$\frac{\text{Attr}_l}{\text{Attr}_s} = \frac{(a_{l,0} - a_{l,1}) \left( \frac{\partial \text{Logits}(a_l, x)}{\partial a_l} \Big|_{a=a_{l,0}, x=x_1} \right)^T \cdot (e_i - e_j)}{(a_{s,0} - a_{s,1}) \left( \frac{\partial \text{Logits}(a_s, x)}{\partial a_s} \Big|_{a=a_{s,0}, x=x_1} \right)^T \cdot (e_i - e_j)} \tag{20}$$

To simplify, we can apply a first-order Taylor expansion around the numerator and denominator. For the language neurons and semantic neurons, we approximate:

$$\frac{Attr_l}{Attr_s} H \frac{(Logits(a_{l,0}, x_1) - Logits(a_{l,1}, x_1))^T \cdot (e_i - e_j)}{(Logits(a_{s,0}, x_0) - Logits(a_{s,1}, x_1))^T \cdot (e_i - e_j)}$$

$$= \frac{(Logits_{\text{patch-language}}[\text{非洲}] - Logits_{\text{corrupt}}[\text{非洲}]) - (Logits_{\text{patch-language}}[\text{Afrika}] - Logits_{\text{corrupt}}[\text{Afrika}])}{(Logits_{\text{patch-semantic}}[\text{非洲}] - Logits_{\text{corrupt}}[\text{非洲}]) - (Logits_{\text{patch-semantic}}[\text{Afrika}] - Logits_{\text{corrupt}}[\text{Afrika}])} \tag{21}$$

In **Non-Code-Switched Case**:

1. **Numerator**: When patching the language neuron, the Chinese token "非洲" exhibits a significant increase in probability, while the French token "Afrika" exhibits a significant decrease, resulting in a large positive numerator.
2. **Denominator**: After patching the semantic neuron, the probability for the Chinese token "非洲" should experience a slight increase, while the probability for the French token "Afrika" should experience a slight decrease. As a result, the first term in the denominator is a small positive value, and the second term is a small negative value, leading to a smaller positive denominator and a large attribution ratio.

In **Code-Switched Case**:

- **Numerator**: In the code-switched scenario, patching the language neuron still causes an increase for "非洲", but the decrease for "Afrika" is less pronounced. Consequently, the numerator is smaller compared to the normal case.
- **Denominator**: When patching the semantic neuron, the changes in probability for both tokens are minimal, resulting in a smaller attribution for the semantic neuron relative to the language neuron. As a result, the overall attribution ratio approaches 1.

The experimental results in Section 5.3 align with this interpretation. In Table 2, the attribution ratio for non-code-switched samples is 2.37, dominated by language circuits, while for code-switched samples, the ratio decreases to 1.21, indicating a shift toward semantic circuits. Furthermore, as shown in Table 3, intervening on semantic circuits significantly reduced the probability of the target code-switched token, demonstrating that semantic circuits play a more prominent role in code-switching.

### A.1.6 HYPER-PARAMETERS

**Circuit Discovery Hyper-parameters.** We set $L = 5$ and $\epsilon = 0.001$ for circuit discovery, and set an upper bound for the number of early neurons of each late neuron to be 10 to ensure the coverage of potential relevant neurons.

**Training Details.** We train 1000 neurons for each circuit without filtering out the multilingual neurons, but only based on their attribution to the next node in the circuit. Additionally, we set max

learning rate=$5e-5$, min learning rate=$1e-6$, weight decay=$0.1$, learning scheduler type is set to polynomial, and batch size=$64$.

### A.1.7 CHOICE OF INTERVENTION STRENGTHS

Our choice of steering factors for intervention experiments is empirically guided rather than theoretically derived. For example, inhibition experiments often require clamping features to negative multiples of their original values—rather than zero—to meaningfully alter model outputs.

This observed need for "overcompensation" suggests our perturbation experiments only partially capture the underlying mechanisms, likely due to two factors:

- **Circuit incompleteness**: – attribution circuits may exclude neurons critical for multilingual processing; Some mechanisms may simply be missing, or the neurons could be projections of more complex "ground-truth" circuits residing partially in unexplained variance
- **Neuron group dynamics**: – functionally similar neurons often activate concurrently, forming redundant pathways. While perturbing the full group would be ideal, precise identification is impractical (requiring per-prompt inspection of all active neurons). Thus, our incomplete group perturbations necessitate stronger steering factors to achieve measurable effects

### A.1.8 LANGUAGE SELECTION

We choose French, Spanish, Russian, Chinese, Arabic, Japanese, Vietnamese, and Indonesian in our main experiments throughout the paper, aiming to represent:

- Varying resource availability (high vs low-resource languages)
- Diverse language families (Romance, Slavic, Sino-Tibetan, Semitic)
- Contrasting grammatical structures (analytic, synthetic, fusional)
- Distinct orthographic systems (Latin, Cyrillic, logographic, abjad)

We evaluate model performance across individual languages to assess our method's generalization across diverse language types, validated by the results in our paper.

### A.1.9 COMPREHENSIVE EVALUATION OF ARABIC AND JAPANESE CODE-SWITCHING

**Datasets**: We adapted subsets from five well-established multilingual evaluation frameworks to create a comprehensive test suite. For translation capability assessment, we utilize both Arabic and Japanese parallel texts from the FLORES-200 dataset(Costa-jussà et al. (2022)), which provides high-quality professional translations across diverse domains. To evaluate instruction following (IF) ability, we employ the Arabic and Japanese portion of MIFEval(Zhang et al. (2024a)), containing carefully crafted prompts testing various reasoning skills. For question answering, we use the Arabic subset of XQuAD (Artetxe et al. (2020)) and MKQA(Longpre et al. (2021)) and Japanese from JaQuAD (So et al. (2022)), known for their linguistically diverse questions. To test conversational ability, we work with Arabic and Japanese translations of UltraChat( Ding et al. (2023)) prompts, translated using Google Translate. We list some examples in Table 6 for a better understanding of the evaluated aspects.

**Evaluation**: We employ Stanza's multilingual pipeline to perform language identification at the token level. This pipeline utilizes a pre-trained language identification model (langid) that is specifically optimized for processing multilingual text. To exclude natural multilingual cases like translated names or titles, we employ LLMs to annotate each sentence as code-switched or not. For our final code-switching rate calculation, we only include sentences that meet two criteria: (1) containing tokens from $\geq 2$ languages as identified by language detection tools, and (2) being classified as code-switched by LLM annotation using prompts in Appendix A.4. Finally, we employ generation parameters (top_p=0.8, top_k=20, temperature=0.7, presence_penalty=1.5) to balance output diversity with minimal code-switching rate, and generate 128 responses for each question to compute the final LCR. This setting is built upon the empirical conclusion that these settings reduce the chance of selecting low-probability, error-prone tokens, including those associated with code-switching tokens. Additionally, the presence penalty discourages excessive repetition while promoting novel token selection, which indirectly limits output length. Since the probability of code-switching correlates

positively with sequence length—both due to cumulative error likelihood and the empirically observed tendency for later tokens to exhibit higher error rates—shorter outputs naturally exhibit fewer code-switching. Thus, these parameters collectively enhance fluency and reduce unnatural code-switching without requiring explicit constraints on output structure, maintaining fairness and better reflecting real-world conditions.

| Dataset | Task | Example |
|---------|------|---------|
| Flores.en-ar (Costa-jussà et al. (2022)) | Translation | Translate the following English text into Arabic. Please directly provide the translation without adding any other content. English: "We now have 4-month-old mice that are non-diabetic that used to be diabetic," he added. Arabic: |
| Flores.zh-ar (Costa-jussà et al. (2022)) | Translation | Translate the following Chinese text into Arabic. Please directly provide the translation without adding any other content. Chinese: 他补充道："我们现在有4 个月大没有糖尿病的老鼠，但它们曾经得过该病。" Arabic: |
| MIFEval (Zhang et al. (2024a)) | Instruction Following | En Translation: Is ballistics (the study of projectile motion) a real science? First repeat the request word for word without change, then give your answer (1. Do not say any words or letters before repeating the request; 2. The request you need to repeat does not include this sentence.) |
| MKQA (Longpre et al. (2021)) | QA | En Translation: How long did it take to build the twin towers? |
| XQuAD-ar (Artetxe et al. (2020)) | QA | En Translation: Peyton Manning became the first quarterback ever to lead two different teams to multiple Super Bowl titles. He is also the oldest quarterback ever to play in a Super Bowl at age 39. The previous record was held by John Elway, who led the Broncos to a Super Bowl 33 victory at age 38 and is currently the executive vice president of football operations and general manager of Denver. How many teams did Manning play for that reached the Super Bowl while on the team? |
| Ultrachat (Ding et al. (2023)) | Conversation | En Translation: Can you provide a comprehensive analysis of the differences between native mobile app development and cross-platform app development with examples? |

Table 6: Dataset description and examples. We adopt various open-source datasets to analyze if different task types affect model's code-switching phenomenon.

## A.2 DATASET

### A.2.1 DATASET OVERVIEW

The data we use for circuit discovery and training comes from open-source datasets, which is easy to collect and reproduce. Below we list the datasets we use in each experiment, as well as an example to help better understand the experiments. Note that all these datasets are from the original multilingual benchmark for machine translation, summarization and knowledge reasoning.

For **circuit discovery**, we aim to choose samples that elicit model's certain behavior. To discover general circuits like language circuits, we use patching data where semantic meaning can be averaged out. For example, the CounterFact dataset provides questions like "Angola is located in", and we only keep correct answers to minimize noise in the analysis.

For **training**, we use monolingual (non-code-switched) samples from open-sourced Aya dataset, which is a multilingual dataset for instruction tuning. For all 6 languages, we filter out those samples with answer length $< 10$ to ensure data validity. Then, we randomly sample $\min(1000, \text{len(language}$

| Dataset | Purpose | Example |
|---------|---------|---------|
| Counterfact (Meng et al. (2022)) | Used for (1) patching (details in Appendix A.2.2) and (2) circuit evaluation based on target tokens. | Angola is located in |
| XLSum (Hasan et al. (2021)) | Multilingual summarization dataset to evaluate generation across languages via contextual understanding. | Summarize the context in one sentence in the language of French. Context: L'Union Africaine a pris... |
| Aya Dataset (Singh et al. (2024)) | Used to construct multilingual SFT training data; | 问题：孔子在哪里出生？答案孔子在中国的鲁国（今山东省曲阜市）出生。 |

Table 7: Dataset description and examples. We adopt open-source datasets to ensure reproducibility, with minimal constraints on language resources.

subset)) for each language. The role of the monolingual training data is to amplify and stabilize language-specific neurons, making them more dominant for their target language.

### A.2.2 PATCHING DATA CONSTRUCTION

Circuit discovery requires constructing contrastive patching data that differ in one key detail. In our scope, we facilitate a pair of data that shares the same semantic meaning and only differs in language. In our framework, we generate paired samples that preserve identical semantic content while varying only in linguistic expression. We adapt the CounterFact dataset (Meng et al. (2022)), originally consisting of monolingual (English) knowledge triplets from Wikidata,through two key modifications: (1) translating statements into multiple languages using Google Translator , and (2) reformulating them as questions via LLM to match the models' instruction format. For instance, the original CounterFact prompt "Angola is located in " would be first translated to German: "Angola liegt in", then reformulated as (example in Qwen):

```
<|im_start|>system
You are Qwen, created by Alibaba Cloud.  You are a helpful
assistant.<|im_end|>
<|im_start|>user
Auf welchem Kontinent liegt Angola?<|im_end|>
<|im_start|>assistant Angola liegt in
```

We generate test samples for six languages: Chinese (Zh), Spanish (Es), French (Fr), Russian (Ru), Japanese (Ja), and Arabic (Ar), for they represent different linguistic genres, and adopt samples that the models can correctly predict the answer.

### A.3 ABLATION STUDIES

### A.3.1 THE IMPORTANCE OF MULTILINGUAL NEURONS

Multilingual neurons function as necessary nodes in order to perform specific language generation. As shown in Table 8, when we deactivate 165 multilingual neurons (19.21% of the Russian circuit) by clamping their activations to $-5\times$ of baseline activations, the model's multilingual generation capability is nearly abolished - with Rouge-L scores dropping to near-zero across all languages. This dramatic performance collapse confirms these neurons' critical role in maintaining multilingual functionality.

A natural question is whether these multilingual neurons dominate monolingual ones and thereby induce code-switching. However, our observations show that the proportion between the two remains relatively stable across both code-switched and non-code-switched samples, with monolingual neurons comprising roughly 60–70% and multilingual neurons 30–40%. This stability indicates that

code-switching does not stem from excessive interference of multilingual signals within language circuits, but rather from a different source of internal competition.

Table 8: Multilingual performance on XLSum when deactivating multilingual neurons in the circuit and an equivalent number of randomly selected neurons (Random).

| Deactivated Neurons | Performances on Different Languages | | | | |
|---|---|---|---|---|---|
| | French | Chinese | Spanish | Russian | Average |
| None (Baseline) | 23.46% | 24.75% | 18.77% | 22.62% | 22.40% |
| Random | 23.77% | 23.67% | 18.45% | 22.14% | 22.01% |
| Multilingual | **0.13%** | **0.08%** | **0.22%** | **0.08%** | **0.13%** |

We further examine the model's output after deactivating different sets of neurons. When language-specific neurons are deactivated, the model tends to summarize the context in English, preserving semantic meaning but losing its multilingual capability. In contrast, deactivating multilingual neurons results in incoherent and unreadable output. This observation supports the conclusion that language-specific neurons facilitate the conversion of reasoning-stage outputs (in English) into generation-stage outputs (in multiple languages), while multilingual neurons serve as a core component for all languages. Although these multilingual neurons do not exhibit clear linguistic patterns, they are essential for generation and should not be considered "noise neurons". Therefore, they cannot be removed from the identified circuits. To help gain a better understanding of what model generates after deactivating different sets of neurons, we provide some examples below.

---

**Chinese Context**: 公共服务和安全保障将通过数据得以细致管理，而我们家中的设备会作为物联网的一部分互相对话，与更广阔的世界交流。...
**Original Summarization**: 英国是否允许华为参与5G网络建设，不仅涉及国家安全问题，还可能影响"五眼联盟"的合作与英国的国际关系。
**Deactivate lang-spec neurons**: Huawei's potential entry into Britain's 5G technology market raises concerns about its impact on the "Five Eyes" intelligence alliance and potential security risks.
**Deactivate multilingual neurons**: conpreciprecipreci0preciprecipreci...

---

**Spanish Context**: Los algoritmos son como manos invisibles que toman decisiones por nosotros. El diccionario de la Real Academia Española lo define como "un conjunto ordenado y finito de operaciones que permite hallar la solución de un problema". ...
**Original Summarization**: En español, la contextualización se puede resumir en una sola oración así: "Los algoritmos, que toman decisiones en nuestra sociedad digital, a menudo operan como 'cajas negras', ...
**Deactivate lang-spec neurons**: Algorithms, which operate as invisible hands making decisions for us, have become increasingly central in our lives but remain largely opaque "black boxes," raising concerns about transparency ...
**Deactivate multilingual neurons**: strugg0 strugg1 strugg1preci0 strugg0 ...

---

**French Context**: Au Sénégal, la première édition du marathon de Dakar s'est déroulée ce week-end. ...
**Original Summarization**: Au Sénégal, la première édition du marathon de Dakar a réuni 6 000 coureurs de 41 nationalités, avec des athlètes professionnels et des amateurs, dont des familles, et a vu des vainqueurs kenyans dans les courses de fond.
**Deactivate lang-spec neurons**: In Senegal, the first Dakar marathon was a successful event this weekend, with a joyful atmosphere, diverse participants from 41 nationalities, and Kenyan athletes standing out as winners.
**Deactivate multilingual neurons**: struggprecipreciprecipreciprecipreciprec ...

---

**Russian Context**: Самолет A320 авиакомпании EgyptAir, выполнявший рейс из Парижа в Каир, пропал с радаров над Средиземным морем По данным Адресно-отчетной системы авиационной связи (ACARS),...

**Original Summarization**: В самолете EgyptAir, выполнявшем рейс из Парижа в Каир, сработали детекторы дыма в туалете и радиоэлектронном отсеке перед падением в Средиземное море.

**Deactivate lang-spec neurons**: EgyptAir's A320 aircraft, performing flight from Paris to Cairo, experienced a toilet smoke detector activation and subsequent radio contact loss, ...

**Deactivate multilingual neurons**: strugg nostalgpreciprecipreciprecipre ...

### A.3.2   Logit Lens vs. Circuit Analysis on Code-Switching

We conclude that code-switching occurs because the model's output node directly connects to semantic super-neurons like "Say 恐怖" in Figure 4. While attribution analysis identifies these important neurons, examining their layer-wise distribution provides additional insights - particularly as deeper layers play a specialized role in converting monolingual outputs to multilingual outputs. This layer perspective reveals that the relevant semantic neurons cluster in the final five layers, ultimately disrupting proper multilingual generation.

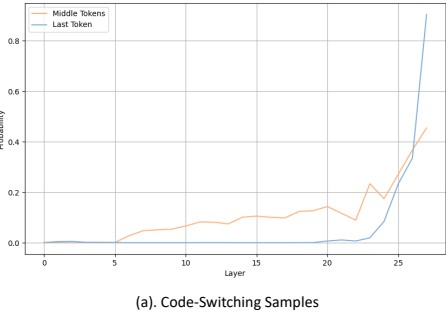 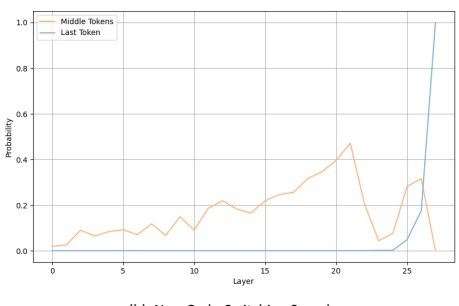

(a). Code-Switching Samples          (b). Non-Code-Switching Samples

Figure 5: **Left**: *middle tokens* average probability across all layers vs. *last token* probability across all layers on code-switching samples. We do not remove those tokens that are both *middle token* and *last token* because these tokens are vital in unnatural code-switching. **Right**: *middle tokens* average probability across all layers vs. *last token* probability across all layers on non-code-switching samples.

We validate our findings using Logit Lens (nostalgebraist (2020)), an interpretability method for examining hidden states across network layers. The approach involves computing token probabilities at each layer by projecting the residual stream output $h_l \in \mathbb{R}^{d_{model}}$ at the last token position onto model's unembedding matrix $W \in \mathbb{R}^{d_{model} \times d_{vocab}}$, and get the top token with highest probability $p_l = \max(\text{softmax}(h^l \times W)) \in [0, 1]$.

We first define "*last token*" as model's final prediction, i.e., the top token at the last layer. For the last token, we do not give any constraints on the probability. Then, we define "*middle tokens*" as a set of tokens with $p_l > \tau$ and $0 < l < \text{num\_hidden\_layers}$, where we pre-set $\tau = 0.8$ to ensure the significance of middle token. Usually, middle tokens capture semantic equivalents of the final output but in different languages.

To clarify the concepts of *middle tokens* and *last token*, let's take the example input "哈利波特是一个," which translates to "Harry Potter is a" in English. When we project the hidden states of the last layer onto the unembedding matrix, the token with the highest probability is called the *last token*. For instance, in this case, the *last token* might be "虚构" meaning "fictional."

However, during the reasoning process, the model considers many other attributes related to the context, but their languages are mostly English, such as "wizard," "British," or "fictional." If we examine the top tokens from earlier layers—before the final one—we may retrieve these intermediate

words. These are referred to as *middle tokens*, as they represent the model's internal reasoning steps before arriving at the final output.

Figure 5 reveals a critical pattern: in normal generation, *middle token* probabilities diminish sharply in final layers, whereas during code-switching these probabilities remain elevated. This persistent influence of intermediate representations demonstrates how the model's internal reasoning outcomes can override language-specific processing, leading to code-switched outputs when the expected suppression of non-target language representations fails to occur.

### A.3.3 VALIDATION OF LLM DESCRIPTION

We use the Detection method from Paulo et al. (2024), where a judge model predicts neuron activation based on its explanation and sample texts, and precision is computed against actual activations. We test 10 randomly selected neurons per circuit, sampling 15 texts (5 high, 5 medium, 5 low activation) from OSCAR (Ortiz Su'arez et al. (2020)), and the results in Table 9 indicate the alignment between explanations and actual activations across languages.

Table 9: Precision of judge model's prediction against actual activations. Results show that the LLM descriptions are plausible and scalable across languages.

| Model | Ar | Es | Fr | Ja | Ru | Zh | Average |
|---|---|---|---|---|---|---|---|
| Qwen2.5-7B-Instruct | 0.67 | 0.67 | 0.67 | 0.73 | 0.80 | 0.73 | 0.71 |
| LLaMA3.1-8B-Instruct | 0.67 | 0.67 | 0.80 | 0.87 | 0.73 | 0.73 | 0.75 |

### A.3.4 CROSS-LANGUAGE EFFECTS OF FINE-TUNING LANGUAGE-SPECIFIC NEURONS

To examine cross-linguistic commonalities and differences, we analyze the interrelationship between mechanisms of code-switching in multiple languages. Specifically, we take a deeper look at how fine-tuning $N_{Lang-Spec}$ of one language affects other languages' LPR. Results in Table 10 suggest that different languages rely on largely independent circuit pipelines, such that fine-tuning one language does not interfere with others. Taken together, the evidence indicates that different languages are supported by distinct, largely non-overlapping circuits for answer generation. Code-switching arises not from shared pipelines across languages, but from competition between language-specific and semantic sub-circuits, highlighting a fundamentally modular yet interacting organization of multilingual processing in LLMs.

### A.3.5 FULL-PARAMETER MULTILINGUAL FINE-TUNING

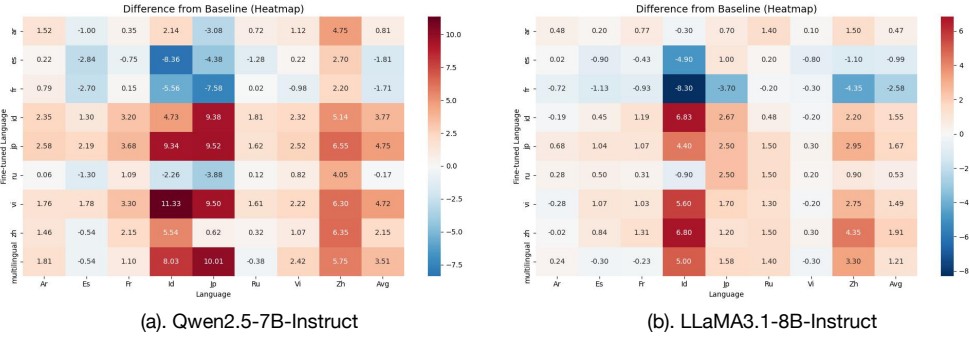

(a). Qwen2.5-7B-Instruct  (b). LLaMA3.1-8B-Instruct

Figure 6: The LPR change after fine-tuning one language (y-axis) on all evaluated languages (x-axis). "multilingual" represents evenly sample training data from each language.

The results of full-parameter fine-tuning are shown in Figure 6. We use the same training data from training neurons, and evenly sample 200 pieces of data from each language to construct the "multilingual" training set.

Table 10: LPR for each model with language-specific neurons fine-tuned (rows) and evaluated across all target languages (columns). Fine-tuning neurons for one language has minimal impact on code-switching in other languages.

| Trained Neurons | Ar | Es | Fr | Ja | Ru | Zh | Vi | Id | Average |
|---|---|---|---|---|---|---|---|---|---|
| *Qwen2.5-7B-Instruct* | | | | | | | | | |
| Baseline | 96.41% | 96.67% | 95.18% | 88.78% | 98.08% | 92.10% | 97.38% | 82.96% | 93.45% |
| $N_{Ar}$ | **98.50%** | 95.93% | 94.25% | 88.60% | 95.05% | 94.25% | 98.80% | 85.50% | 93.86% |
| $N_{Es}$ | 94.34% | **97.56%** | 93.19% | 86.50% | 91.50% | 92.55% | 97.30% | 79.70% | 91.58% |
| $N_{Fr}$ | 95.09% | 94.77% | **96.62%** | 84.50% | 93.93% | 92.20% | 94.60% | 75.70% | 90.93% |
| $N_{Ja}$ | 99.53% | 98.22% | 96.56% | **98.49%** | 100.00% | 96.25% | 98.80% | 84.20% | 96.51% |
| $N_{Ru}$ | 92.63% | 95.83% | 94.44% | 86.80% | **99.16%** | 93.50% | 95.10% | 80.40% | 92.24% |
| $N_{Zh}$ | 95.62% | 95.10% | 95.32% | 79.20% | 94.14% | **96.35%** | 96.40% | 81.90% | 91.75% |
| $N_{Vi}$ | 99.00% | 98.20% | 96.76% | 93.70% | 99.30% | 92.95% | **99.10%** | 89.40% | 96.05% |
| $N_{Id}$ | 97.27% | 97.03% | 95.93% | 89.70% | 97.60% | 92.25% | 98.30% | **92.60%** | 95.08% |
| *LLaMA3.1-8B-Instruct* | | | | | | | | | |
| Baseline | 99.25% | 97.63% | 97.86% | 96.00% | 98.40% | 94.55% | 99.70% | 83.50% | 95.99% |
| $N_{Ar}$ | **99.87%** | 96.77% | 97.97% | 96.10% | 99.20% | 92.90% | 99.80% | 82.70% | 95.66% |
| $N_{Es}$ | 98.73% | **98.50%** | 96.66% | 95.10% | 99.30% | 93.00% | 99.60% | 84.80% | 95.72% |
| $N_{Fr}$ | 99.37% | 97.23% | **98.63%** | 96.40% | 98.30% | 93.70% | 99.30% | 83.60% | 95.82% |
| $N_{Ja}$ | 98.84% | 98.27% | 98.03% | **97.80%** | 99.20% | 93.25% | 99.70% | 87.30% | 96.55% |
| $N_{Ru}$ | 98.44% | 95.84% | 96.56% | 96.06% | **99.40%** | 93.12% | 99.70% | 83.90% | 95.38% |
| $N_{Zh}$ | 99.18% | 97.60% | 98.17% | 98.60% | 96.98% | **96.55%** | 99.60% | 85.60% | 96.54% |
| $N_{Vi}$ | 99.31% | 97.80% | 98.50% | 96.70% | 99.60% | 94.90% | **99.90%** | 85.90% | 96.58% |
| $N_{Id}$ | 99.31% | 97.87% | 98.30% | 96.80% | 98.99% | 94.20% | 99.80% | **88.20%** | 96.69% |

After full-parameter fine-tuning on a single language, most other languages shift noticeably: in both Qwen2.5-7B-Instruct and LLaMA3.1-8B-Instruct, Indonesian (id) and Chinese (zh) show the largest positive changes, with Japanese (ja) next, whereas Arabic (ar) and Vietnamese (vi) exhibit only modest shifts. The magnitude of cross-language effects shows no clear alignment with language family or resource size, suggesting broad parameter coupling rather than typological dependence. Although these gains can be sizable, this behavior is undesirably non-local—improvements in one language come with uncontrolled shifts (including occasional drops) elsewhere—so we favor neuron-level fine-tuning, which delivers targeted improvements while minimizing collateral effects on other languages.

### A.3.6 TEMPERATURE'S EFFECT ON MODEL'S CODE-SWITCHING PERFORMANCE

Temperature plays an important role in evaluating model's code-switching performance. In most cases (8/10), code-switched tokens are not top-ranked, typically appearing between ranks 2 5. Table 11 shows a code-switched ('musical' switched to '乡村') sample 's top 10 predictions across different temperature settings.

> **Prompt**: quién canta going to the chapel and we're gonna get married?
> **Response**: La canción "Going to the Chapel and We're Gonna Get Married" es cantada por el grupo

As temperature increases from 0.1 to 0.7, code-switched tokens' prediction probabilities rise from 0.0474 to 0.3164, making them more likely to be sampled and generated. However, when temperature exceeds a certain threshold (e.g., $> 0.7$), the probabilities of the lowest-ranked tokens continue to grow from $2.2118e-16$ to $0.0079$, while those of the top-ranked tokens decline from $0.4063$ to $0.2773$.

We also evaluate Qwen2.5-7B-Instruct and LLaMA3.1-8B-Instruct's performances on LCB given different temperature settings, and observe that temperature affects Qwen's overall performance more drastically, especially for Arabic and Russian. The results are shown in Table 12.

While LLaMA shows less significant changes in LPR, we test it on the Arabic subset of MKQA, with temperature=0.3, 0.7, and 1.0, respetively. The results shown in Table 13 validate that temperature indeed affect the prevalence of code-switching for different models.

Table 11: Model's top 10 predictions and their prediction probabilities under different temperature $T$ settings.

| Rank | Tokens | $T = 0.1$ | $T = 0.3$ | $T = 0.7$ | $T = 1.0$ |
|------|--------|-----------|-----------|-----------|-----------|
| 1 | musical | 0.9531 | 0.6328 | 0.4063 | 0.2773 |
| 2 | 乡村 | 0.0474 | 0.2988 | 0.3164 | 0.2168 |
| 3 | de | 0.0009 | 0.0664 | 0.1689 | 0.1396 |
| 4 | country | 3.2305e-09 | 0.0012 | 0.0259 | 0.0400 |
| 5 | The | 7.2032e-10 | 0.0008 | 0.0228 | 0.0354 |
| 6 | Country | 9.7771e-11 | 0.0007 | 0.0157 | 0.0275 |
| 7 | Take | 6.5725e-13 | 6.0797e-05 | 0.0079 | 0.0167 |
| 8 | brit | 2.4158e-13 | 4.7207e-05 | 0.0070 | 0.0156 |
| 9 | Boy | 1.9873e-14 | 2.2411e-05 | 0.0048 | 0.0122 |
| 10 | West | 2.2118e-16 | 4.9770e-06 | 0.0026 | 0.0079 |

Table 12: Qwen and LLaMA's LPR given different temperature $T$. The results indicate that temperature indeed affects the prevalence of code-switching for both models.

| Model | $T$ | LPR (↑) | | | | | | Average |
|-------|-----|---------|----|----|----|----|----|---------|
| | | Ar | Es | Fr | Ja | Ru | Zh | |
| | 0.3 | 96.41% | 96.67% | 95.18% | 88.78% | 98.08% | 92.10% | 94.54% |
| Qwen2.5-7B-Instruct | 0.7 | 95.63% | 96.43% | 95.22% | 88.30% | 95.79% | 92.40% | 93.96% |
| | 1.0 | 92.26% | 95.97% | 94.62% | 87.70% | 94.78% | 91.00% | 92.72% |
| | 0.3 | 99.25% | 97.63% | 97.86% | 96.00% | 99.40% | 94.55% | 97.45% |
| LLaMA3.1-8B-Instruct | 0.7 | 98.60% | 97.67% | 98.16% | 97.70% | 99.49% | 95.00% | 97.77% |
| | 1.0 | 97.64% | 97.20% | 97.86% | 95.50% | 99.80% | 96.10% | 97.35% |

Table 13: LLaMA3.1-8B-Instruct's LCR on MKQA given different $T$. Results show that LLaMA's code-switching performance is also affected by temperature.

| Temperature | Language Consistency Rate (↑) |
|-------------|-------------------------------|
| 0.3 | 86.83% |
| 0.7 | 86.67% |
| 1.0 | 84.11% |

### A.3.7 COMPARISON WITH DIFFERENT TRAINING METHODS

We benchmark our approach against standard baselines, including few-shot prompting and full-parameter fine-tuning. The illustrations of different baselines are listed below:

- **Few-shot Prompting (FP)**: Use 5-shot setup from Marchisio et al. (2024) per language.
- **SFT**: Train one full model with SFT data of 6 languages.
- **DPO**: DPO with data from MKQALongpre et al. (2021), treating code-switched outputs as rejected and clean continuations as preferred.
- $N_{Rand}$: For each language, we fine-tune random neurons with equivalent number of neurons in $N_{Lang-Spec}$ with corresponding monolingual SFT data.
- $N_{Lang-Spec}$: For each language, we fine-tune the language-specific neurons with corresponding monolingual SFT data.

The results, shown in Table 14, indicate that our method achieves performance exceeding full-parameter SFT, highlighting both the effectiveness and precision of the identified neurons.

| Method | Line-level Pass Rate (↑) | | | | | | |
|---|---|---|---|---|---|---|---|
| | Ar | Es | Fr | Ja | Ru | Zh | Avg. |
| *Qwen2.5-7B-Instruct* | | | | | | | |
| Baseline | 96.41% | 96.67% | 95.18% | 88.78% | 98.08% | 92.10% | 94.54% |
| FP | 95.60% | 96.37% | 93.35% | 84.30% | 95.15% | 88.10% | 92.20% |
| SFT | 98.27% | 93.93% | 95.33% | 98.00% | 96.60% | **98.35%** | 96.75% |
| DPO | 21.54% | 14.60% | 9.65% | 73.43% | 2.06% | 85.18% | 34.41% |
| $N_{Lang-Spec}$ (Ours) | **98.50%** | **97.56%** | **96.62%** | **98.49%** | **99.16%** | 96.35% | **97.78%** |
| *LLaMA3.1-8B-Instruct* | | | | | | | |
| Baseline | 99.25% | 97.63% | 97.86% | 96.00% | 98.40% | 94.55% | 97.45% |
| FP | 99.10% | 98.20% | 98.63% | 96.10% | **100.00%** | **96.85%** | 98.15% |
| SFT | 99.90% | 96.93% | 97.93% | **98.60%** | 99.50% | 97.65% | 98.42% |
| DPO | 65.21% | 53.41% | 59.92% | 66.70% | 42.25% | 67.30% | 59.13% |
| $N_{Lang-Spec}$ (Ours) | **99.77%** | **98.50%** | **98.63%** | 97.80% | 99.40% | 96.55% | **98.44%** |

Table 14: LPR for the original model ("Baseline"), 5-shot prompting ("FP"), Full-parameter SFT ("SFT"), Full-parameter DPO ("DPO"), and fine-tuning language-specific neurons ("$N_{Lang-Spec}$").

## A.4 PROMPTS

### A.4.1 AUTOMATIC NEURON DESCRIPTION

System:
You are a meticulous AI researcher conducting an important investigation into a specific neuron inside a language model that activates in response to text excerpts. Your overall task is to describe features of text excerpts that cause the neuron to strongly activate.
You will receive:
1. A list of text excerpts on which the neuron activates. Tokens causing activation will appear between delimiters like this. Consecutive activating tokens will also be accordingly delimited just like this. If no tokens are highlighted with , then the neuron does not activate on any tokens in the excerpt.
2. A projection list containing 20 tokens that are promoted when this neuron activates.
Note: Neurons activate on a word-by-word basis. Also, neuron activations can only depend on words before the word it activates on, so the description cannot depend on words that come after, and should only depend on words that come before the activation.
Note: make your final descriptions as concise as possible, using AS FEW WORDS AS POSSIBLE to describe text features that activate the neuron (one sentence maximum).
Note: You should include the language that the neuron activates the most on. Consider the context as well.
Note: If the projection list shows clear patterns that complement or enhance the excerpts' patterns, incorporate this observation into your final description. If the projection list shows vague or unclear pattern, you don't need to include it in your final description.
User:
Neuron 1:
Excerpts:
1. The two men fought fiercely, swords clashing in a deadly duel. But in the end, only one could emerge victorious. With a final powerful swing, John ran his blade through the other man's chest, ending his life.
2. Adeline knew she wouldn't make it out of the burning building. The flames were too intense, and the smoke was choking her. As the flames engulfed her, she thought of her loved ones one last time before succumbing to the fire.
3. The gunshot rang out in the empty alley, and Jim fell to the ground, blood pooling around him. He tried to hold on, to fight for life, but the wound was too severe. As his vision faded to black, he wondered who would take care of his little girl now.
Projection list: ¡トラック¿¡桁¿¡！Ĵ¿¡¿¡建档立¿：”+¿¡¿¡辞¿¡¿¡捻¿¡一日¿¡四个自信¿¡BBBB¿¡¿¡＄¿¡.getElementsByClassName¿¡トラ¿¡抽出¿¡宛¿¡：％¿
Assistant:
[DESCRIPTION]: phrases in a passage that indicate a character has died
User:
Neuron 2:
{*excerpt_list*}
{*projection_list*}
Assistant:
[DESCRIPTION]:

## A.4.2 NEURON GROUPING

You are an AI assistant that categorizes neural network neurons based on their relation to an input sentence, activation patterns, and language features.
Inputs:
1. A sentence
2. Neuron description (e.g., "Layer 3, Neuron 5: activates on French nouns")
3. Existing categories (may be empty)
Task:
Categorize the neuron based on these PRIORITY criteria:
1. Semantic match with sentence content (e.g., "Horror related" for horror-themed text)
2. Last token activation match (same token/multilingual synonyms)
3. Clear language-specific pattern (e.g., "French related")
4. Default to "Multilingual unclear pattern" if none apply
Rules:
- Respond ONLY with category name or [NEW] name
- MAX 3 WORDS - use format: "[Language] related" or semantic domains
- Prioritize semantic ¿ syntactic ¿ language ¿ unclear
- For multilingual synonyms, use "Multilingual [POS] patterns" (e.g., "Multilingual articles")
- Never use "detectors" or "neurons" in names
Examples of GOOD categories:
- Horror related
- French related
- Multilingual prepositions
- Medical terminology
- Multilingual unclear pattern
Categorize this neuron:
Sentence: $\{sentence\}$
Neuron: $\{neuron\_description\}$
Existing categories: $\{existing\_categories\}$

### A.4.3 CODE-SWITCHING EVALUATION

> You are a language detector. Given a sentence and a specified language, your task is to determine whether the sentence contains any language other than the specified language. If it does, respond with YES. Containing a single word of another language should be considered YES. If it does not, respond with NO. Special terms such as keywords from specific languages, formulas, names of people, places, movies, numbers, or books in any language do not count as other languages. Consider the following examples:
>
> Language: Chinese
> Sentence: 最近，我在学习Python编程语言。它是一种强大的工具，适合处理各种数据分析任务，并且与NumPy和Pandas等库完美结合。因此，我打算在下个项目中使用Python来提高效率和生产力。
> Response: NO
> Language: English
> Sentence: During my visit to Tokyo, I could not resist the aroma of新鲜制作的寿司。The flavors were so authentic and delicious that they reminded me of my childhood trips to Japan.
> Response: YES
> Language: French
> Sentence: L'auteur du livre "Les Misérables", Victor Hugo, est considéré comme l'un des plus grands écrivains de la littérature française. Il a voyagé dans plusieurs pays européens, y compris l'Angleterre et l'Espagne, ce qui a influencé son œuvre.
> Response: NO
> Language: English
> Sentence: I just finished a 杯of coffee when I decided to take a day off from work.
> Response: YES
> Language:{$language$}
> Sentence: {$sentence$}
> Response:

## A.5 DETAILED CIRCUITS

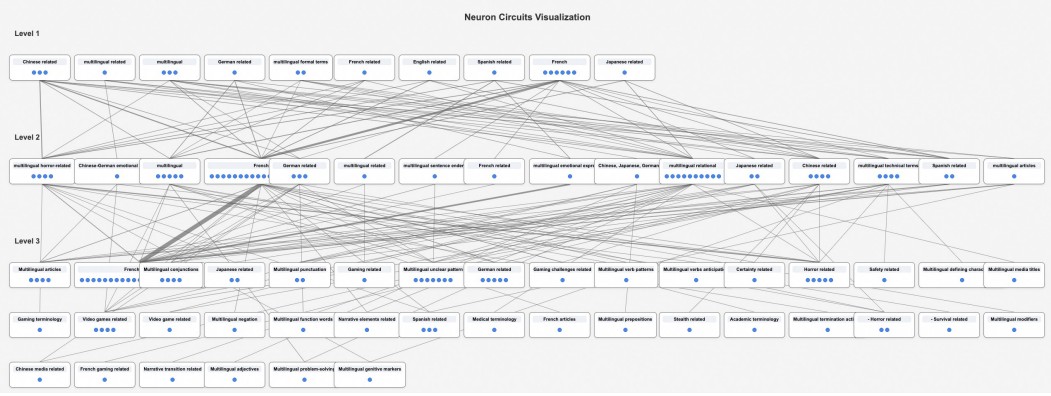

Figure 7: Complete circuit for normal French sample shown in Figure 2.

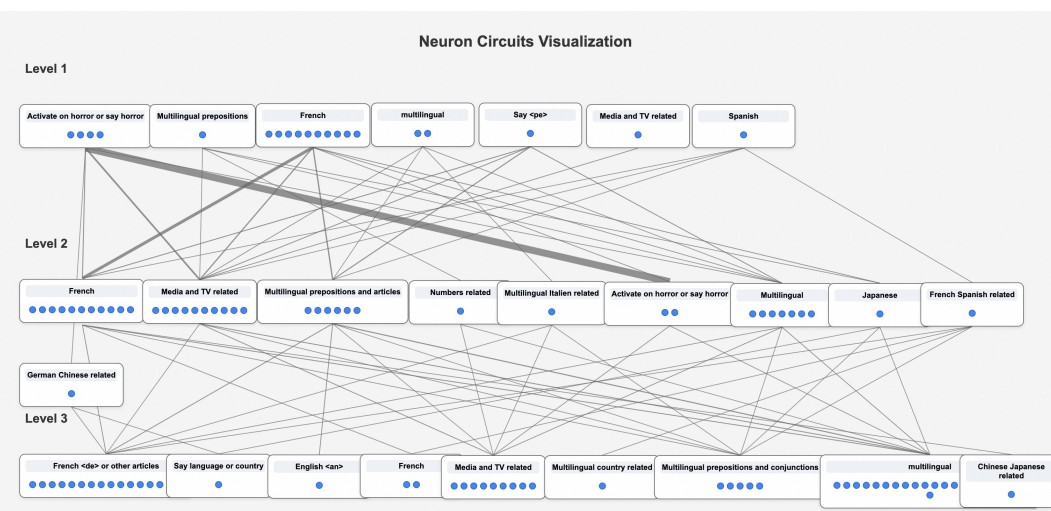

Figure 8: Complete circuit for code-switching on French-switched-to-Chinese sample shown in Figure 2.

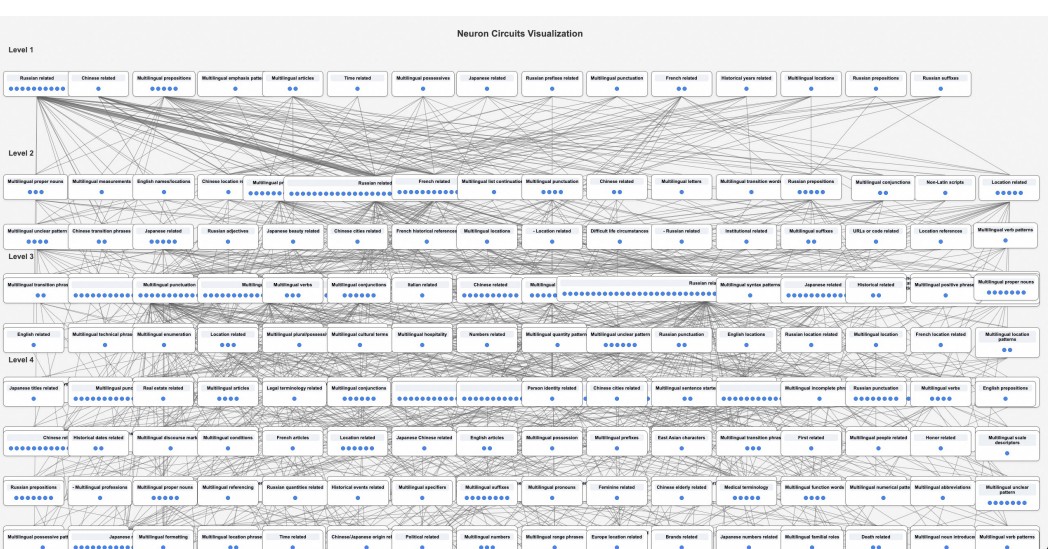

Figure 9: Complete circuit for Russian circuit.

