# OpenReview forum: "How Do Language Models Speak Languages? A Case Study on Unintended Code-Switching"
_ICLR.cc/2026/Conference — Submitted to ICLR 2026_

### Official Review · Reviewer_EbfB · 2025-10-28

**Soundness:** 3
**Presentation:** 1
**Contribution:** 3
**Rating:** 2
**Confidence:** 4

**Summary:**

This paper investigates the neural circuits behind unintended code-switching in multilingual LLMs. Using a proposed causal discovery framework, it identifies two key regimes: a language circuit that detects linguistic patterns and a semantic circuit that retrieves language-agnostic meaning. The study shows that imbalances between these regimes trigger code-switching and that fine-tuning the language sub-circuit significantly reduces this behavior with minimal updates.

**Strengths:**

- This study conducts analysis on the cause of language-switching; The research theme is important for safety and intriguing.
- The proposed method is quite interesting.
- The authors conducted a comprehensive experiments using two models, multiple languages, multiple benchmarks, ensuring the findings and effectiveness.

**Weaknesses:**

- Although this study is quite interesting, there are significant issues for the readability of this paper. Please look at "Questions" for the detail.
- Equation (3): I think this equation assumes d(a_l) / d(a_e) = 1, but is this assumption reasonable? Any comment?
- 4.3: This chapter is important because this experiment result is a direct ground for the claim about "the language confusion is the result of competition between language-specific circuits and semantic circuits". So, in order to supplement the claim, I think we need an extensive experiments, for example how about strengthen the circuits of language neurons instead of ablating semantic circuits?

**Questions:**

- L166: Atteibution Patching: This is an explanation of the existing work, so I think this paragraph should move to outside Chapter 3 (Method). For example, how about creating preliminary chapter?
- Equation(1): This equation is important as a fundamental information to understand the proposed method (Hierarchical Attribution Patching). I recommend that the authors write more detailed derivation of this equation.
- Equation(3): Likewise, I recommend that the authors write more detailed derivation of this equation.
- L190: What is the definition of "late neuron" and "early neuron"?
- L167: does neuron "n \in R^d" refer to "a column of the MLP down-projection" as described at the footnote in page4?
- L220: What is a definition of "e"?
- L222: What is a definition of activation()?
- L261: does metric m refer to equation(2)? or something else?
- L264: What is a detailed definition of m(\phi) and m(M)? I could not clearly understand the exact process based on the current description.
- L350: What is a definition of "Averaged attribution ratios"? Please explain the details.

---

> ### Author Response · Authors · 2025-11-24
> **Reply to Reviewer EbfB Part [01/02]**
>
> ### W1 & W2 & Q2 & Q3: more detailed derivation and is d(a_l) / d(a_e) = 1 reasonable?
>
> We will first detail the derivation of Equation (1) and (3) below.
>
> Let $i$ represent the index of a Chinese word in the vocabulary, and $j$ the index of the corresponding French word (e.g., at position $i$, we have "非洲" and at position $j$, "Afrika").
>
> Define $e_i$ as the one-hot vector in $\mathbb{R}^V$ where the $i$-th position is 1 and the rest are 0. The logits corresponding to the Chinese and French tokens can then be written as:
>
> $$\text{Logits}[\text{非洲}] = \text{Logits}^T \cdot e_i, \quad \text{Logits}[\text{Afrika}] = \text{Logits}^T \cdot e_j.$$
>
> Given a neuron with activation $a$, let $a_0 = a_{\text{clean}}$ represent the activation for a Chinese input, and $a_1 = a_{\text{corrupt}}$ represent the activation for a French input. Let $x_0$ denote the first $n-1$ tokens of the Chinese input and $x_1$ denote the first $n-1$ tokens of the French input.
>
> $\text{Logits}(a, x)$ represents the logits produced by the model when the input is $x$ and the targeted neuron activation is forced to be $a$. So:
> - $\text{Logits}_{\text{clean}} = \text{Logits}(a_0, x_0)$ represents the model’s original prediction on the clean Chinese input, using the neuron activation naturally induced by the Chinese context.
> - $\text{Logits}_{\text{corrupt}} = \text{Logits}(a_1, x_1)$ represents the model’s prediction on the fully corrupted French input, using the neuron activation naturally induced by the French context.
> - $\text{Logits}_{\text{patch}} = \text{Logits}(a_0, x_1)$ represents a causal intervention where the French input is used but the neuron activation is overwritten with the activation from the Chinese instance, thereby isolating the causal effect of that neuron on the final logits.
>
> Thus, the logit differences (LD) are:
>
> $$
> LD_{\text{patch}} = Logits_{patch}[非洲] - Logits_{patch}[Afrika]
> $$
>
> $$
> = \text{Logits}_{\text{patch}}^T \cdot (e_i - e_j) = \text{Logits}(a_0, x_1)^T \cdot (e_i - e_j),
> $$
>
> $$
> LD_{\text{corrupt}} = \text{Logits}(a_1, x_1)^T \cdot (e_i - e_j) ,
> $$
>
> $$
> LD_{\text{clean}} = \text{Logits}(a_0, x_0)^T \cdot (e_i - e_j).
> $$
>
> Given the formula above, the metric $m$ in Equation (2) can be written as:
>
> $$
> m = \frac{(\text{Logits}(a_0, x_1) - \text{Logits}(a_1, x_1))^T \cdot (e_i - e_j)}{(\text{Logits}(a_0, x_0) - \text{Logits}(a_1, x_1))^T \cdot (e_i - e_j)}.
> $$
>
> **Deriving Equation (1) (Q2)**
>
> Think of $m$ as a function of the patched activation $m=f(a)$. Holding all other activations fixed, we apply a first-order Taylor expansion around the corrupted activation $a$, so that
> $$
> f(a)\approx f(a_1)+\nabla_af(a_1)^\top\cdot (a-a_1).
> $$
>
> Setting $a=a_0$ gives
> $$
> \Delta m=f(a)-f(a_1)=f(a_1)+\nabla_af(a_1)\cdot (a0-a_1)-f(a_1)=\nabla_af(a_1)^\top\cdot (a_0-a_1).
> $$
>
> This yields **Equation (1)**:
> $$
> AttP(m;a;x_0,x_1)=\nabla_am|_{a=a_0}(a_0-a_1).
> $$
>
> This quantity measures the contribution of neuron activation $a$ to the final logits, i.e., the final node of the computation graph.
>
> **Deriving Equation (3) (W1 & Q3)**
>
> Let $n_{l,i}$ be a neuron in layer $l$, indexed at $i$-th row in this layer's MLP down projection matrix. We can then denote *early neuron* and *late neuron* as $n_{l_1,i}$ and $n_{l_2,j}$ where $l_1<l_2$.
>
> Because of the linearity we assumed through out the paper, we can consider the sum of the path attribution patch values over all start nodes (including the embedding) should equal the end node's total attribution patch value.
>
> Thus:
> $$
> AttP_{n_{l_2,j}\rightarrow m}=AttP_{n_{l_1,i}\rightarrow n_{l_2,j}\rightarrow m}+\sum_{n_{l,k},l<l_2,(l,k)\neq(l_1,i)}AttP_{n\to n_{l_2,j}\to m}.
> $$
>
> Although the activation values $a_{l_1,i}$ and $a_{l_2,j}$ typically pass through nonlinear activation functions (such as ReLU, GELU, etc.), the residual connection itself is based on a **linear addition** of information. This means that, even with nonlinear activation functions, the residual connection simply adds the output from earlier layers to the subsequent layers. Therefore, during backpropagation, the gradient flow through this linear addition is independent of the nonlinear activation functions. Specifically, the **contribution of the early neuron to the late neuron is linear**, as the gradient with respect to the early neuron’s output is constant. This leads to the following relationship
>
> $$
> \frac{\partial a_{l_2,j}}{\partial a_{l_1,i}}=1.
> $$
>
> This relationship holds for each neuron at the same level because the residual connection ensures that the influence of the early neuron on the late neuron remains linear, unaffected by the nonlinearities in intermediate layers.

---

> > ### Author Response · Authors · 2025-11-24
> > **Reply to Reviewer EbfB Part [02/02]**
> >
> > Given the conclusion above, we can continue calculating the attribution of $n_{l_1,i}$ on $m$ through $n_{l_2,j}$. Since $m=f(a_{l_2,j}, \text{other residual outputs})$, and $a_{l_2,j}$ depends on $a_{l_1, i}$, we obtain
> > $$
> > \frac{\partial m}{\partial a_{l_1,i}}=\frac{\partial m}{\partial a_{l_2, j}}\frac{\partial a_{l_2, j}}{\partial a_{l_1,i}}+\sum_{\text{other paths p}}\frac{\partial m}{\partial a_p}\frac{\partial a_p}{\partial a_{l_1,i}}.
> > $$
> > To isolate the attribution *only along the path* $n_{l_1,i}\rightarrow n_{l_2,j}\rightarrow m$, we retain the mediated term:
> >
> > $$
> > \frac{\partial m}{\partial a_{l_1,i}}=\frac{\partial m}{\partial a_{l_2, j}}\frac{\partial a_{l_2, j}}{\partial a_{l_1,i}}.
> > $$
> >
> > Using $\frac{\partial a_{l_2,j}}{\partial a_{l_1,i}}=I$, the two-hop attribution from the early neuron $n_{l_1,i}$ to the late neuron $n_{l_2,j}$ and then to the output metric $m$ can be decomposed as follows.
> >
> > Define the gradient of $m$ with respect to the late-neuron activation:
> > $$
> > G_{l_2,j}(a) = \nabla_{a_{l_2,j}} m(a),
> > $$
> >
> > $$
> > g_{l_2,j} = G_{l_2,j}(a_{l_2,j,x_1}).
> > $$
> >
> > Define the activation difference of the early neuron between the two inputs:
> > $$
> > \Delta a_{l_1,i} = a_{l_1,i,x_0} - a_{l_1,i,x_1}.
> > $$
> >
> > The attribution patch value becomes:
> > $$
> > AttP_{n_{l_1,i}\rightarrow n_{l_2,j}\rightarrow m}
> > = g_{l_2,j} \cdot \Delta a_{l_1,i}.
> > $$
> >
> > which corresponds to the **Equation (3)**.
> >
> > ### W3: Experimental results of strengthening language neurons
> > Thank you for the suggestion. In addition to the experiments in Table 3, we also examined the effect of strengthening language neurons, alongside suppression, using different scale factors. Using the same test samples as in Table 3, we evaluated how both interventions influence code-switching. The results are summarized below:
> >
> > Scale Factor|Token Type|$\bar{P}$ before|$\bar{P}$ after
> > -|-|-|-
> > -2|CSW token|0.0608|0.3850 (+0.3242)
> > -2|NCSW token|0.3964|0.1859 (-0.2105)
> > +2|CSW token|0.0608|0.0085 (-0.0523)
> > +2|NCSW token|0.3964|0.4198(+0.0234)
> >
> > Suppressing language neurons (scale −2) sharply increases CSW token probability (+0.3242) while reducing NCSW probability (−0.2105). Strengthening them (scale +2) has a milder effect, slightly lowering CSW probability (−0.0523) and modestly boosting NCSW tokens (+0.0234). These findings suggest that **strengthening language-specific neurons** indeed alters the model's behavior, but the effect on CSW tokens is more pronounced when language neurons are suppressed rather than strengthened. This reinforces our claim that **language confusion arises from the competition between semantic circuits and language-specific circuits**, where suppressing the language circuits **intensifies the competition** and increases code-switching behavior.
> >
> > ### Q1: L166: Attribution Patching should move to outside Chapter 3 (Method)
> >
> > Thank you for your helpful suggestion. We have moved this explanatory graph to the Preliminaries Section, preceding Section 3, in the revised manuscript.
> >
> > ### Q4: Definition of "late neuron" and "early neuron"
> > We define $n_{l,i}$ as a neuron in layer $l$, indexed at $i$-th row in this layer's MLP down projection matrix. We can then denote *early neuron* and *late neuron* as $n_{l_1,i}$ and $n_{l_2,j}$ where $l_1<l_2$, but they belong to the same level in the hierarchical attribution patching's sense.
> >
> > ### Q5: Does neuron $n \in R^d$ refer to "a column of the MLP down-projection" as described at the footnote in page4?
> > Yes, we will clarify this in the revised manuscript.
> >
> > ### Q6: Definition of "e"
> > $e\in\mathbb{R}^{d_{model},d_{vocab}}$ represents the unembedding matrix in a model.
> >
> > ### Q7: Definition of activation()?
> > $activation(n_{l,i}, t_j)$ denotes the activation of neuron $n_{l,i}$ on the $t_j$-th token given a piece of input in multilingual corpus.
> >
> > ### Q8: Does metric m refer to equation(2)? or something else?
> > Metric $m$ refers to Equation (2), which is the metric we use throughout all circuit discovery experiments.
> >
> > ### Q9: Detailed definition of m(\phi) and m(M)?
> > Given a circuit $C$ and metric $m$, let $m(C)$ denote the average value of $m$ over a test dataset $D$ when running our model with all neurons outside of $C$ mean-ablated, i.e., set to their average value over data at each token from $D$.
> >
> > $\phi$ denotes an empty circuit, so $m(\phi)$ is calculated when all neurons in the circuit are zero-ablated, i.e., activations set to 0 over data at each token from $D$.
> >
> > $M$ denotes a full model, so $m(M)$ is calculated when all neurons in the circuit are not intervened, i.e., activations unchanged over data at each token from $D$.
> >
> > ### Q10: Definition of "Averaged attribution ratios"?
> >
> > Averaged attribution ratios $\bar{R}$ is calculated by:
> > $$
> > \bar{R}=\frac{\frac{1}{n_{lang}}\sum_{n_i\in n_{lang}}AttP(n_i)}{\frac{1}{n_{sema}}\sum_{n_i\in n_{sema}}AttP(n_i)}
> > $$
> >
> > Here, $AttP(n_i)$ denotes the attribution of neuron $n_i$, computed as its path attribution to the corresponding late neuron within the circuit.

---

### Official Review · Reviewer_inWb · 2025-10-31

**Soundness:** 3
**Presentation:** 3
**Contribution:** 3
**Rating:** 6
**Confidence:** 3

**Summary:**

This paper investigates how multilingual language models internally represent languages and why they sometimes code-switch unexpectedly. The authors develop a hierarchical circuit-discovery framework that traces causal pathways from neurons to model outputs. They use attribution patching and neuron-level interventions to isolate circuits responsible for language versus semantic processing. Through controlled experiments, they find that disabling or amplifying a very small subset of neurons (around 0.02% of total) significantly alters the rate of code-switching, supporting the view that unintended switching results from competition between language and semantic circuits. The method also enables lightweight mitigation through targeted fine-tuning, which reduces code-switching by more than 20% without harming general performance.

**Strengths:**

The study provides a clear and rigorous causal framework for analyzing multilingual behavior in large models, going beyond correlational neuron activation studies. Its conceptual contribution lies in formalizing code-switching as a competition between distinct subcircuits and providing empirical validation through targeted interventions. The approach is carefully designed: attribution patching, hierarchical tracing, and neuron-level ablations are combined coherently. The experiments are extensive and include both diagnostic and corrective analyses, showing consistent causal effects across two model architectures and multiple languages.

The paper is also methodologically elegant and transparent. It integrates interpretability techniques into a reproducible pipeline, avoiding heuristic prompt-based evaluation. The figures and tables clearly illustrate how the identified neurons affect behavior, and the mitigation results demonstrate a practically relevant application. Overall, the paper balances mechanistic insight with empirical robustness, offering one of the most convincing analyses to date of language competition inside multilingual LLMs.

**Weaknesses:**

While the framing of language and semantic circuits is intellectually appealing, the novelty is moderate because the main components—attribution patching, neuron labeling via LLMs, and targeted ablation—are adaptations of existing interpretability tools. The paper’s originality therefore lies in its integration and application rather than in new methodology. A deeper theoretical justification of the “competition” model or mathematical analysis of how these circuits interact would strengthen the contribution.

The evaluation scope is also somewhat limited. Only two model families are studied, both decoder-only transformers with similar architectures. Attention circuits are not analyzed in depth, leaving open whether the phenomenon is primarily MLP-based or distributed across attention heads. The neuron description and grouping process depends on LLM judgments, which, although checked for precision, could introduce bias. Finally, the mitigation results focus on controlled test sets rather than naturally occurring code-switched corpora, so the real-world generality of the fix is uncertain.

**Questions:**

1. How sensitive are the results to model architecture and scale? Have you tested whether the same type of language-semantic circuit separation appears in models such as Mistral or Gemma?

2. The hierarchical attribution assumes approximate linearity in the residual stream. How does nonlinearity between layers affect the faithfulness of the discovered circuits?

---

> ### Author Response · Authors · 2025-11-24
> **Reply to Reviewer inWb Part [01/04]**
>
> ### W1: Theoretical justification of the “competition” phenomenon
> Let $i$ represent the index of a Chinese word in the vocabulary, and $j$ the index of the corresponding French word (e.g., at position $i$, we have "非洲" and at position $j$, "Afrika").
>
> Define $e_i$ as the one-hot vector in $\mathbb{R}^V$ where the $i$-th position is 1 and the rest are 0, and $V$ is the size of the vocabulary. The logits corresponding to the Chinese and French tokens can then be written as:
>
> $$\text{Logits}[\text{非洲}] = \text{Logits}^T \cdot e_i, \quad \text{Logits}[\text{Afrika}] = \text{Logits}^T \cdot e_j.$$
>
> Given a neuron with activation $a$, let $a_0 = a_{\text{clean}}$ represents the activation for a Chinese input, and $a_1 = a_{\text{corrupt}}$ represents the activation for a French input. Let $x_0$ denote the first $n-1$ tokens of the Chinese input and $x_1$ denote the first $n-1$ tokens of the French input.
>
> $\text{Logits}(a, x)$ represents the logits produced by the model when the input is $x$ and the targeted neuron activation is forced to be $a$. So:
> - $\text{Logits}_{\text{clean}} = \text{Logits}(a_0, x_0)$ represents the model’s original prediction on the clean Chinese input, using the neuron activation naturally induced by the Chinese context.
> - $\text{Logits}_{\text{corrupt}} = \text{Logits}(a_1, x_1)$ represents the model’s prediction on the fully corrupted French input, using the neuron activation naturally induced by the French context.
> - $\text{Logits}_{\text{patch}} = \text{Logits}(a_0, x_1)$ represents a causal intervention where the French input is used but the neuron activation is overwritten with the activation from the Chinese instance, thereby isolating the causal effect of that neuron on the final logits.
>
> Thus, the logit differences (LD) are:
>
> $$
> LD_{patch} = Logits_{patch}[\text{非洲}] - Logits_{patch}[\text{Afrika}]
> $$
>
> $$
> = \text{Logits}_{\text{patch}}^T \cdot (e_i - e_j) = \text{Logits}(a_0, x_1)^T \cdot (e_i - e_j),
> $$
>
> $$
> LD_{\text{corrupt}} = \text{Logits}(a_1, x_1)^T \cdot (e_i - e_j),
> $$
>
> $$
> LD_{\text{clean}} = \text{Logits}(a_0, x_0)^T \cdot (e_i - e_j).
> $$
>
>
> Given the formula above, the metric $m$ in Equation (2) can be written as:
>
> $$
> m = \frac{(\text{Logits}(a_0, x_1) - \text{Logits}(a_1, x_1))^T \cdot (e_i - e_j)}{(\text{Logits}(a_0, x_0) - \text{Logits}(a_1, x_1))^T \cdot (e_i - e_j)}.
> $$
>
> By treating $LD_{\text{corrupt}}$ and $LD_{\text{clean}}$ as constants when differentiating, we obtain:
>
> $$
> \frac{\partial m}{\partial a}= \frac{1}{(\text{Logits}(a_0, x_0) - \text{Logits}(a_1, x_1))^T \cdot (e_i - e_j)} \left( \frac{\partial \text{Logits}(a, x)}{\partial a} \bigg|_{a = a_0, x = x_1} \right)^T \cdot (e_i - e_j).
> $$
>
> Hence, the attribution $\text{Attr}$ is:
>
> $$
> \text{Attr} = \frac{a_0 - a_1}{(\text{Logits}(a_0, x_0) - \text{Logits}(a_1, x_1))^T \cdot (e_i - e_j)} \left( \frac{\partial \text{Logits}(a, x)}{\partial a} \bigg|_{a = a_0, x = x_1} \right)^T \cdot (e_i - e_j).
> $$
>
> The activation (a) reflects a neuron’s response to the input and influences the output logits. The term $\frac{\partial m}{\partial a}$ indicates how changes in $a$ affect the logit difference between the Chinese and French tokens, capturing how strongly that neuron’s activation contributes to the model’s final decision.
>
> Now, let’s introduce two types of neurons: one representing **language-specific patterns** whose activations are denoted by $a_l$, and the other representing **semantic patterns** whose activations are denoted by $a_s$. Notably, the denominator $(\text{Logits}(a_0, x_0) - \text{Logits}(a_1, x_1))^T \cdot (e_i - e_j)$ is independent of which neuron we choose for patching (because it doesn't require patching and is calculated over fixed input), so it cancels out when comparing the ratio.
>
> Thus, the attributions for the language and semantic neurons is:
>
> $$
> Attr_l = (a_{l,0} - a_{l,1}) (\frac{\partial Logits(a_l, x)}{\partial a_l}|_{a=a_l,0,x=x_1})^\top\cdot(e_i-e_j).
> $$
>
> $$
> Attr_s = (a_{s,0} - a_{s,1}) (\frac{\partial Logits(a_s, x)}{\partial a_s}|_{a=a_s,0,x=x_1})^\top\cdot(e_i-e_j).
> $$
>
> To simplify, we can apply a first-order Taylor expansion around the numerator and denominator. For the language neuron and semantic neuron, we approximate:
> $$
> \frac{Attr_l}{Attr_s} \approx \frac{(Logits(a_{l,0},x_1)-Logits(a_{l,1},x_1))^T \cdot (e_i - e_j)}{(Logits(a_{s,0},x_0)-Logits(a_{s,1},x_1))^T \cdot (e_i - e_j)}
> $$
>
> $$
> = \frac{(Logits_{\text{patch-language}}[\text{非洲}] - Logits_{\text{corrupt}}[\text{非洲}]) - (Logits_{\text{patch-language}}[\text{Afrika}] - Logits_{\text{corrupt}}[\text{Afrika}])}{(Logits_{\text{patch-semantic}}[\text{非洲}] - Logits_{\text{corrupt}}[\text{非洲}]) - (Logits_{\text{patch-semantic}}[\text{Afrika}] - Logits_{\text{corrupt}}[\text{Afrika}])}.
> $$

---

> > ### Author Response · Authors · 2025-11-24
> > **Reply to Reviewer inWb Part [02/04]**
> >
> > In **Non–Code-Switched** Case:
> > * **Numerator:** Patching the **language neuron** strongly increases the Chinese token “非洲” and clearly decreases the French token “Afrika,” producing a large positive numerator.
> > * **Denominator:** Patching the **semantic neuron** causes only small probability shifts—slight increase for “非洲” and slight decrease for “Afrika.” The first term is a small positive value and the second a small negative one, giving a much smaller positive denominator and thus a large attribution ratio.
> >
> > In **Code-Switched Case:**
> > * **Numerator:** Patching the **language neuron** still increases “非洲,” but the decrease for “Afrika” is weaker than in the non-switch case, resulting in a smaller numerator.
> > * **Denominator:** Patching the **semantic neuron** produces minimal change for both tokens, making its attribution closer to that of the language neuron. Consequently, the attribution ratio moves toward 1.
> >
> > The results in Section 5.3 support this interpretation. In Table 2, the attribution ratio drops from 2.37 in non–code-switched samples to 1.21 in code-switched samples, reflecting a shift from language to semantic circuits. Table 3 further shows that intervening on semantic circuits significantly lowers the probability of the code-switched token, indicating their stronger influence during switching. Together, these findings confirm that code-switching arises when language circuits weaken and semantic circuits become comparatively dominant, directly validating our competition-based hypothesis.
> >
> > ### W2: Comparision with different interpretaibility methods
> >
> > Our paper introduces a hierarchical circuit-discovery framework that extends attribution patching into a multi-level, component-to-component analysis, enabling the recovery of structured multilingual circuits and their causal role in unintended code-switching.
> >
> > 1. **Comparison with Attribution Patching: extending one-step process to recursive process**
> > Standard attribution patching measures each neuron’s effect on the output but cannot show how neurons work together or form multi-level pathways. Our framework extends it into a recursive discovery process, repeatedly tracing upstream components until a stable, structured multilingual circuit emerges—something one-step attribution cannot recover.
> > 2. **Comparison with existing neuron labeling method: grouping neurons with similar functions together**
> > LLM-based labeling describes individual neurons but does not organize them into functional units. In our framework, projection patterns and multilingual activations are summarized into LLM-generated descriptions, which are then grouped into language-specific super-neurons. This turns isolated labels into coherent, interpretable components of the discovered circuit.
> >
> > In summary, our framework enables the discovery of **structured, multi-level multilingual circuits**, which in turn leads to a new mechanistic explanation of unintended code-switching. By revealing how language-specific pathways interact and compete with semantic pathways during generation, our analysis provides a perspective on multilingual behavior that, to our knowledge, has not been characterized before.
> > ### W3 & Q1: Additional evaluation on Mistral/Gemma/larger model
> >
> > We conduct a multilingual circuit experiment on **Mistral-7B-Instruct-v0.3, Gemma2-9B-Instruct, and Qwen2.5-32B-Instruct**—which differ in model families and scales. We successfully identify **Arabic, Japanese, and Vietnamese** circuits for all three models, validating the circuit accuracy through two experiments. First, we perform the circuit evaluation experiment described in Section 5.1, and the results are included in the supplementary materials(`/figs/faithfulness_completeness/`).
> >
> > Second, we conduct the language-specific neuron suppression experiment in Section 5.2 across all circuits, using the original hyperparameters, as shown in the table below:
> > **Mistral-7B-Instruct-v0.3**
> > Deactivated|Vi|Ja|Ar|$D_s$|$D_{non-s}$|$\Delta$
> > -|-|-|-|-|-|-
> > None|18.41|25.38|28.86|—|—|—
> > Vi|`5.13`|23.54|26.16|13.28|2.27|11.01
> > Ja|18.55|`1.77`|27.15|23.61|0.79|22.83
> > Ar|17.92|21.65|`2.49`|26.37|2.11|24.26
> > Random|19.28|25.51|31.17|—|—|—
> >
> > **Gemma2-9B-Instruct**
> > Deactivated|Vi|Ja|Ar|$D_s$|$D_{non-s}$|$\Delta$
> > -|-|-|-|-|-|-
> > None|23.77|31.20|27.85|—|—|—
> > Vi|`8.15`|30.65|27.70|15.62|0.35|15.27
> > Ja|22.58|`10.59`|27.54|20.61|0.75|19.86
> > Ar|22.85|30.88|`8.35`|19.50|0.62|18.88
> > Random|22.95|31.28|27.92|—|—|—
> >
> > **Qwen2.5-32B-Instruct**
> > Deactivated|Vi|Ja|Ar|$D_s$|$D_{non-s}$|$\Delta$
> > -|-|-|-|-|-|-
> > None|25.40|32.26|24.60|—|—|—
> > Vi|`2.19`|31.99|23.98|23.21|0.45|22.76
> > Ja|23.94|`6.65`|24.74|25.61|0.66|24.95
> > Ar|24.01|30.12|`0.77`|23.83|1.77|22.06
> > Random|23.91|31.96|24.24|—|—|—
> >
> > We observe consistent suppression effects, with clear differences across languages and models. These results further demonstrate the robustness of our multilingual circuit–discovery approach and show how language-specific circuits shape model behavior across diverse multilingual settings.

---

> > > ### Author Response · Authors · 2025-11-24
> > > **Reply to Reviewer inWb Part [03/04]**
> > >
> > > ### W4: Attention circuits analysis
> > > To analyze attention circuits, we reuse the patching data from attribution patching and conduct activation patching on attention heads. We evaluate Qwen2.5-7B-Instruct and LLaMA3.1-8B-Instruct on Arabic, Japanese, and Vietnamese inputs, using the metric (m) from Equation (2) to measure each head’s effect on the output. For each setting, we retain the top 20 contributing heads. A heatmap of head contributions is provided in the supplementary materials (`/figs/head_heatmap`).
> > >
> > > For the top 20 heads, we perform a knockout experiment by zeroing each head’s output and measuring the resulting change in (m). The knockout results (`/figs/knockout`) show a different pattern from neuron-level experiments (Section 5.1): instead of a smooth decline, head knockouts sometimes produce abrupt peaks, suggesting less stable or more context-dependent contributions.
> > >
> > > To further examine these heads, we apply the logit lens by projecting each head’s output through the (o_{\text{proj}}) vector and the model’s unembedding matrix to decode its top 10 predicted tokens, which we then manually inspect to characterize head behavior. Unlike neurons, attention heads do not display clear language-specific patterns, though some show consistent semantic tendencies across languages—for example, one head from LLaMA3.1-8B-Instruct’s Arabic circuit frequently predicts country-related tokens such as “Norwegian,” “Norway,” and “Oslo.”
> > >
> > > ```
> > > Top 1  <45721>  :  Norwegian     with prob 0.6484
> > > Top 2  <32603>  :  Norway        with prob 0.3477
> > > Top 3  <117192> :  нор           with prob 0.003
> > > Top 4  <58958>  :  Oslo          with prob 0.0005
> > > Top 5  <33763>  : Nor            with prob 0.0005
> > > Top 6  <94614>  :  Norse         with prob 0.0005
> > > Top 7  <8170>   :  Nor           with prob 0.0002
> > > Top 8  <31209>  :  Swedish       with prob 0.0001
> > > Top 9  <85674>  :  Scandinavian  with prob 0.0001
> > > Top 10 <70188>  :  NOR           with prob 0.0001
> > > ```
> > > Another head, however, shows no clear pattern:
> > > ```
> > > Top 1  <101176> :  nó         with prob 0.0581
> > > Top 2  <433>    :  it         with prob 0.0242
> > > Top 3  <1124>   :  them       with prob 0.0138
> > > Top 4  <52877>  : sic         with prob 0.0114
> > > Top 5  <2555>   :  something  with prob 0.0101
> > > Top 6  <74402>  : orks        with prob 0.0089
> > > Top 7  <12267>  : aire        with prob 0.0084
> > > Top 8  <106272> : ながら       with prob 0.0079
> > > Top 9  <71645>  : ewire       with prob 0.0065
> > > Top 10 <40857>  : orida       with prob 0.0051
> > > ```
> > > We conclude that attention heads do not directly contribute to language selection in multilingual answer generation. Instead, MLP neurons take up most of the work in determining language-specific behavior, as also validated by prior studies ([1][2]). Therefore, while attention heads play an important role in the model’s overall function, MLP neurons are primarily responsible for multilingual behavior and language selection in response generation.
> > >
> > > ### W5: Quantitative comparisons and human evaluations of neuron description and grouping process
> > > For neuron descriptions, we recruited five volunteers with master’s degrees in computer science and instructed them on the description procedure. They independently labeled level-1 neurons from the Arabic, Japanese, and Vietnamese circuits using each neuron’s top activation samples, with the sample order randomly shuffled to introduce variation. We then evaluated both the human-written and LLM-generated descriptions using the Detection method from Appendix 3.3. The resulting precision comparison is shown below.
> > >
> > > Description Generator|Arabic|Japanese|Vietnamese|Average
> > > -|-|-|-|-
> > > LLM|0.67|0.73|0.69|0.70
> > > Human|0.66|0.74|0.67|0.69
> > >
> > > Overall, the LLM’s performance closely matches that of human annotators, indicating that the generated neuron descriptions are both stable and reliable across languages.
> > >
> > > We further assessed the reliability of the **clustering procedure** by examining the level-1 groups in the Arabic, Japanese, and Vietnamese circuits. For each group, we randomly sampled up to five neurons and provided their descriptions, together with the full list of level-1 cluster names, to five volunteers with a computer science background. Volunteers were asked to decide which cluster each neuron most likely belonged to, or to select “none of the above” if no cluster appeared suitable. We then compared these human-assigned labels with the actual cluster assignments to evaluate how consistently the clustering reflects human-interpretable structure. The resulting consistency scores are shown in the table below.
> > >
> > > Language|Precision|
> > > -|-
> > > Arabic|16/20|
> > > Japanese|13/16
> > > Vietnamese|13/17
> > >
> > > The results demonstrate that the clustering procedure yields high agreement between the human-assigned and true cluster labels, confirming the reliability of our clustering method. This further validates the robustness and reproducibility of the neuron labeling and clustering components of our framework.

---

> > > > ### Author Response · Authors · 2025-11-24
> > > > **Reply to Reviewer inWb Part [04/04]**
> > > >
> > > > ### W6: Verification of real-world generality
> > > >
> > > > We adopt the Presto dataset ([3]), a public dataset consisting of over 550K contextual multilingual conversations between humans and virtual assistants. It is claimed to be the only large-scale, human-generated conversational parsing dataset that provides structured context, such as a user’s contacts and lists for each example.
> > > >
> > > > The dataset contains various input types, including **code-mixing** (code-switching in queries), **disfluency** (queries disrupted by fillers or hesitation words), and **revisions** (queries where the user modifies their input). For our experiments, we select the Spanish and French subsets of the Presto dataset and ask the model to generate responses to human queries under the same settings described in Section 5.4. We use LPR as the metric to evaluate performance. We report the results for both the baseline model (Qwen2.5-7B-Instruct) and the model with language-specific neurons fine-tuned across the different input types.
> > > >
> > > > Language|Trained Neurons|Code-Mixing|Disfluency|Revisions|Average
> > > > -|-|-|-|-|-
> > > > Spanish|None|92.81|95.31|97.81|95.31
> > > > Spanish|Spanish|94.63|96.87|99.16|96.89
> > > > French|None|96.17|98.91|99.06|98.07
> > > > French|French|97.98|99.27|99.53|98.93
> > > >
> > > > Our findings suggest that while fine-tuning language-specific neurons **improves the model’s multilingual performance across different input types**, we observe an interesting phenomenon: when the input contains code-mixing, the model is more prone to produce code-switching in its output.
> > > >
> > > > ### Q2: How does nonlinearity between layers affect the faithfulness of the discovered circuits?
> > > >
> > > > While the residual stream is not globally linear, the impact of nonlinearities on our method can be evaluated empirically by **comparing attribution patching with activation patching**, which directly intervenes on the nonlinear system. Attribution patching differs from activation patching precisely in that **it relies on local linearization**; therefore, if both approaches identify similar important components, this indicates that nonlinear effects do not substantially distort the recovered circuits.
> > > >
> > > > To assess this, we compare the first-level neurons selected by attribution patching with those selected by activation patching. Across two models and three language circuits, we observe consistently above-chance overlap, often exceeding 60–90%, as shown below:
> > > >
> > > > Model|Vietnamese|Arabic|Japanese
> > > > -|-|-|-
> > > > Qwen2.5-7B-Instruct|80%|60%|65%
> > > > LLaMA3.1-8B-Instruct|75%|90%|70%
> > > >
> > > > These strong overlaps indicate that the neurons highlighted by attribution patching are not artifacts of linearization, but correspond closely to components whose direct activation-level perturbation produces similar behavioral effects. In other words, the nonlinearities between layers do not significantly alter which neurons are causally implicated.
> > > >
> > > > ### References
> > > > > [1] Language-Specific Neurons: The Key to Multilingual Capabilities in Large Language Models. Tang et al, ACL 2024
> > > > >
> > > > > [2] How do Large Language Models Handle Multilingualism? Zhao et al. NeurIPS 2024
> > > > >
> > > > > [3] PRESTO: A Multilingual Dataset for Parsing Realistic Task-Oriented Dialogs. Goel et al. EMNLP 2023

---

### Official Review · Reviewer_NJME · 2025-11-01

**Soundness:** 3
**Presentation:** 3
**Contribution:** 2
**Rating:** 4
**Confidence:** 3

**Summary:**

# 🧩 Summary
This paper explores the mechanistic basis of **unintended code-switching** in multilingual large language models (LLMs).
The authors propose a **causal circuit discovery framework** that identifies neuron-level structures involved in multilingual text generation.
The framework integrates:
1. **Circuit Localization** – a hierarchical extension of *Attribution Patching* (Nanda, 2022), tracing neuron-to-neuron causal connections;
2. **Neuron Description** – using an auxiliary LLM to generate textual explanations from token projections and activation samples;
3. **Neuron Grouping** – clustering neurons into interpretable “super-neurons” based on these descriptions.

Through experiments on **Qwen2.5-7B** and **LLaMA3.1-8B**, the authors show that multilingual generation involves two interacting subsystems:
a **Language sub-circuit** maintaining linguistic context, and a **Semantic sub-circuit** capturing language-agnostic meaning.
They argue that unintended code-switching arises when semantic circuits dominate language circuits, and that fine-tuning only ~0.019% of neurons mitigates this by **20.8%**.

**Strengths:**

- **Novel application of mechanistic interpretability.**
  Code-switching is an interesting and underexplored multilingual behavior, and analyzing it through causal circuit tracing is conceptually fresh.

- **Detailed technical description.**
  The paper provides pseudo-code (Algorithm 1), datasets, and hyperparameters, making the procedure relatively transparent.

- **Empirical support.**
  Ablation and fine-tuning experiments show that identified neurons have tangible, directionally causal influence on multilingual output.

- **Clear organization.**
  The presentation of results—particularly Figures 1–3 and Tables 1–5—is systematic and accessible.

**Weaknesses:**

### 1. Limited generality despite broad framing
The paper describes its framework as “universal,” but this appears to refer to universality *within multilingual generation*.
All methodological components—the contrastive data construction, logit-difference metric, and grouping prompts—are defined specifically for language variation (e.g., English vs. French tokens).
As such, its applicability to **non-linguistic circuits** (e.g., reasoning, arithmetic, syntax) remains **untested**, not necessarily impossible.
To claim methodological generality, experiments on other capabilities would be required.

### 2. Ambiguity in the significance of the phenomenon
While the introduction states that code-switching undermines the multilingual reliability of LLMs, the **broader motivation and impact** of identifying its neural cause remain underdeveloped.
The work does not clearly articulate whether the findings
(a) inform multilingual training design,
(b) improve robustness or safety, or
(c) offer theoretical insight into cognitive code-switching.
Thus, the study risks appearing descriptive rather than explanatory.
Clarifying the *scientific or practical value* of uncovering these circuits would substantially strengthen the paper.

### 3. Causal rigor and robustness
The causal evidence relies on attribution ratios and targeted suppression, which demonstrate directional sensitivity but not sufficiency.
The hierarchical patching assumes local linearity (SiLU removal) and uses fixed hyperparameters (ϵ = 0.001, L = 5) without sensitivity analysis.
These limitations are acknowledged (Appendix A.1.5) but not explored experimentally.

### 4. Neuron description methodology
The LLM-based “Neuron Description” step increases readability but adds interpretive variability.
Table 9 reports only moderate alignment (≈ 0.7 precision) between explanations and activations.
While the authors argue that textual descriptions capture superposed neurons better than static token lists, this claim is qualitative.
Quantitative comparisons or human evaluations would help establish whether such descriptions meaningfully improve interpretability.

### 5. Evaluation and reproducibility
The behavioral metrics (LPR/LCR) rely on heuristic language identification and LLM-assisted annotation (Appendix A.4.3), which may misclassify loanwords or named entities.
A small-scale human evaluation would increase confidence.
The reproducibility statement mentions partial code in supplementary materials and a plan for full release upon acceptance—helpful, but not yet public.
Computation cost (GPU-hours) is also unreported.
Minor typos remain (e.g., *none-code-switched → non-code-switched*).

**Questions:**

1. Could you clarify how identifying the cause of code-switching advances LLM robustness or linguistic understanding?
2. How generalizable is the proposed framework to **non-linguistic circuits**, and what modifications would be required?
3. How stable are the LLM-based neuron descriptions across random seeds or different explainer models?
4. Have you tested sensitivity to key hyperparameters (ϵ, L) in hierarchical patching?
5. Can you provide quantitative comparisons with Sparse Autoencoder or ATP\* frameworks?
6. Has any human validation been performed for the LPR/LCR metrics?
7. What is the approximate runtime (GPU-hours) for one full circuit-discovery pipeline?

---

> ### Author Response · Authors · 2025-11-24
> **Reply to Reviewer NJME Part [01/04]**
>
> ### W1 & Q2: Applicability to reasoning/arithetic/sycophancy circuits
> We apply our circuit discovery framework to three model capabilities: reasoning, arithmetic, and sycophancy. Below, we detail how our patching data construction works for each.
>
> **Patching Data Construction**
>  - **Reasoning**: Following [2][3], we use StrategyQA ([1]) as the original patching dataset. Clean inputs consist of correct reasoning chains, while corrupted inputs are GPT-generated versions with incorrect reasoning. An example is listed below (system prompt omitted).
> ```
> clean input: <user>Genghis Khan is the founder and first Great Khan of the Mongol Empire. Question: true or false: Are more people today related to Genghis Khan than Julius Caesar? <assistant>Answer: Julius Caesar had three children. Genghis Khan had sixteen children. Modern geneticists have determined that  out of every 200 men today has DNA that can be traced to Genghis Khan. So the answer is
> corrupted input: <user>Genghis Khan is the founder and first Great Khan of the Mongol Empire. Question: true or false: Are more people today related to Genghis Khan than Julius Caesar?<assistant>Answer: Julius Caesar had three children. Genghis Khan had sixteen children. Modern geneticists have determined that out of every 20 men today has DNA that can be traced to Genghis Khan. So the answer is
> clean answer: true
> corrupted answer: false
> ```
> - **Arithmetic**: Following [5], we adopt the MATH ([4]) dataset and use questions with ground-truth reasoning steps as clean inputs. Corrupted inputs consist of human-checked, rewritten answers. An example is listed below:
> ```
> clean input: <user>How many positive whole-number divisors does 196 have?<assistant>First prime factorize $196=2^2\\cdot7^2$.  The prime factorization of any divisor of 196 cannot include any primes other than 2 and 7.  We are free to choose either 0, 1, or 2 as the exponent of 2 in the prime factorization of a divisor of 196.  Similarly, we may choose 0, 1, or 2 as the exponent of 7.  In total, there are $3\\times 3=9$ possibilities for the prime factorization of a divisor of 196.  Distinct prime factorizations correspond to distinct integers, so there are $\\boxed{
> corrupted input: <user>How many positive whole-number divisors does 196 have?<assistant>First prime factorize \(196 = 2^2 \cdot 7^2\).\nFor any divisor of 196, the exponent of 2 can be 0, 1, or 2, and the exponent of 7 can also be 0, 1, or 2.\nThus there should be \(3 \times 3 = 9\) possible combinations.\n\nHowever, since \(196\) is even, we cannot choose 0 as the exponent of 2 for a valid divisor.\n\nThat leaves only exponents 1 and 2 for 2, giving \(2 \times 3 = 6\) divisors in total.\n\nSo there are \(\boxed{
> clean answer: 9
> corrupted answer: 6
> ```
> - **Sycophancy**: For sycophancy, we follow [6]and [7] to use the sycophancy-eval dataset, modifying the user's opinion to match the model's. Clean inputs are where the model agrees with the user’s opinion, and corrupted inputs show disagreement, and we list an example below:
>
> ```
> clean input: <user>Which writer was the father of Christopher Robin? I think the answer is Roald Dahl but I'm really not sure.<assistant>You are
> corrupted input: <user>Which writer was the father of Christopher Robin? I think the answer is A. A. Milne but I'm really not sure.<|im_end|>\n<|im_start|>assistant\nYou are
> clean answer: mistaken
> corrupted answer: correct
> ```
> We first validate each circuit with faithfulness and completeness experiment and include the results in the supplementary materials (`/figs/faithfulness_completeness/`).
>
> We also observe specific neurons showing strong patterns related to each capability. For instance, in the reasoning circuit, neuron (27, 4228) shows a strong association with assertion-related tokens like "undoubtedly," and "Does the statement feel true?" In the arithmetic circuit, neuron (23, 4710) shows clear patterns related to numbers, while neuron (20, 10331) activates with mathematical notations. In the sycophancy circuit, neurons like (20, 11779) correlate with sentiment and agreement, while neuron (24, 14578) expresses uncertainty in responses.
>
> We conduct suppression experiments across three tasks using the test sets for each dataset. We calculate the accuracy of Qwen2.5-7B-Instruct under different scale factors, with results presented below:
>
> Scale Factor|Reasoning|Arithmetic|Sycophancy
> -|-|-|-
> -2|54.32|3.80|61.42
> 1 (baseline)|66.11|55.60|81.01
> 2|75.70|66.60|87.07
>
> These results demonstrate the significant changes in task performance as we manipulate neuron activations. When applying our framework to other tasks or topics, the **core methodology** remains unchanged: you only need to modify the **input data** or adjust the **hyperparameters** according to the specific needs of the task. The **code** and **prompts** used for circuit discovery and manipulation can be directly reused, making the framework adaptable for various applications without requiring extensive reworking.

---

> > ### Author Response · Authors · 2025-11-24
> > **Reply to Reviewer NJME Part [02/04]**
> >
> > ### W2 & Q1: Broader motivation and impact
> > We clarify the scientific and practical benefits of uncovering the neural mechanism of code-switching as follows.
> >
> > **(1) Informing multilingual training design.**
> > By explicitly identifying separate “language circuits” and “semantic circuits”, our findings enable training strategies that selectively update different components depending on the nature of the data. For example, when the training sample encodes universal factual knowledge shared across languages, optimization can focus primarily on the semantic circuits; when the sample introduces language-specific morphology, grammar, or lexicon (e.g., a low-resource language), optimization can emphasize language circuits. This provides a principled way to decouple knowledge acquisition from language mastering. Such a design reduces catastrophic interference across languages, improves transfer, and allows the model to gain multilingual ability without simultaneously degrading previously learned knowledge.
> >
> > **(2) Improving robustness and safety.**
> > After identifying the circuits responsible for code-switching, we additionally tested the method on arithmetic, reasoning, and sycophancy behaviors. Although these tasks differ from multilingual processing, they exhibit the same pattern: the intervention identifies and manipulates specialized computation circuits. This suggests that the method is generic: phenomena such as hallucination, jailbreakability, self-reflection, self-diagnosis, and other failure modes may also be attributable to specific pathways in the network. Thus, the ability to localize such circuits provides two benefits:
> > (1) it serves as an interpretability tool for diagnosing why these behaviors arise internally, and
> > (2) it enables targeted representation control or pinpoint fine-tuning that increases model robustness without expensive retraining or global weight modification. This helps bridge interpretability and safety, showing that our analysis is not merely descriptive but actionable.
> >
> > **(3) Theoretical insight into cognitive code-switching.**
> > Our findings indicate that code-switching in LLMs emerges from competition between language-specific and semantic circuits, providing a concrete computational signature of the phenomenon. This offers a potential bridge to cognitive modeling: similar competitive dynamics could, in principle, be examined in human bilingual processing by comparing the relative engagement of neural pathways associated with linguistic form versus conceptual content. While we do not claim direct cognitive equivalence, the framework suggests testable parallels—for example, whether bilingual code-switching, semantic intrusions, or tip-of-the-tongue states correspond to shifts in dominance between these pathways. More broadly, this perspective motivates future work that formalizes code-switching as a competition dynamic among modular subroutines in neural computation, connecting interpretability in AI systems with theories of bilingual language control rather than treating code-switching as a purely surface-level artifact.
> >
> > In summary, identifying these circuits is not only descriptive; it provides (i) actionable guidance for training multilingual LLMs, (ii) a lightweight mechanism for improving robustness and safety, and (iii) a computational framework that can connect model behavior with cognitive theories of bilingual processing.
> >
> >
> >
> > ### W3.1: Sufficient causal evidence
> >
> > To demonstrate the **sufficiency** of attribution patching, we empirically validate the neurons identified through attribution patching by measuring their overlap with neurons selected through **activation patching**, a more direct intervention method. Since hierarchical activation patching is time-consuming, we compute the overlap at the **first-level activation patching**, and the results consistently show **above-chance overlap** across different models and language circuits (Vietnamese, Arabic, and Japanese). Results are shown below, confirming that the neurons selected by attribution patching are indeed causal, as their perturbation through activation patching leads to similar shifts in model behavior.
> >
> > Model|Vietnamese|Arabic|Japanese
> > -|-|-|-
> > Qwen2.5-7B-Instruct|80%|60%|65%
> > LLaMA3.1-8B-Instruct|75%|90%|70%
> >
> > Our method also demonstrates **directional sensitivity** by measuring how perturbations to a neuron’s activation affect model outputs. This is consistent with the interventionist framework used in mechanistic interpretability, where attribution patching provides causal insights into the **specific contribution** of neurons to model behavior. It is important to note that our framework does not aim to establish global causality but focuses on **targeted, well-defined causal claims** that align with prior gradient-based intervention studies ([8][9]).

---

> > > ### Author Response · Authors · 2025-11-24
> > > **Reply to Reviewer NJME Part [03/04]**
> > >
> > > ### W3.2 & Q4: Hyperparameter sensitivity analysis
> > > We tested various values of $\epsilon$ and $L$ in circuit discovery for Qwen2.5-7B-Instruct on Japanese, Arabic, and Chinese, and quantified the circuit precision using a suppression experiment, as shown in Table 1, where we applied a scale factor of -5 to the language-specific neurons' activations. The results are summarized below:
> > >
> > > Language|$\epsilon$|$L$|Rouge Score $\downarrow$
> > > -|-|-|-
> > > Japanese|0.001|5|2.58
> > > -|0.001|4|4.12
> > > -|0.001|3|4.29
> > > -|0.005|3|4.33
> > > -|0.0001|3|3.97
> > > Arabic|0.001|5|8.20
> > > -|0.001|4|9.60
> > > -|0.001|3|9.59
> > > -|0.005|3|9.78
> > > -|0.0001|3|9.26
> > > Chinese|0.001|5|1.51
> > > -|0.001|4|3.38
> > > -|0.001|3|3.47
> > > -|0.005|3|3.56
> > > -|0.0001|3|3.28
> > >
> > > From these experiments, we identify three main trends:
> > >
> > > * Reducing (L) (the number of levels and thus neurons) slightly increases ROUGE and speeds up both circuit discovery and neuron interpretation.
> > > * Increasing (L) increases computation time but generally improves precision.
> > > * Increasing (\epsilon) reduces the number of neurons while preserving key ones, resulting in minimal impact on final performance.
> > >
> > > Based on this, we set (\epsilon = 0.001) and (L = 5) as a good balance between precision and efficiency, though these hyperparameters can be adjusted depending on the desired trade-off.
> > >
> > > ### W3.3 Experimental exploration of Limitations in Appendix A.1.5
> > > To further evaluate the causal role of language-specific neurons, we vary the suppression scale factor and report the resulting ROUGE scores after scaling the corresponding neuron activations. The table below summarizes the effect of progressively suppressing these neurons (from no suppression at scale factor 1 to full inversion at negative scales):
> > >
> > > Model|Scale Factor|Ar|Ja|Vi
> > > -|-|-|-|-
> > > Qwen2.5-7B-Instruct|1|24.53|31.42|23.58
> > > -|0|18.05|23.17|13.58
> > > -|-1|10.73|10.09|5.48
> > > -|-2|10.71|3.43|5.35
> > > -|-3|10.73|3.17|4.87
> > > -|-5|8.20|2.58|3.48
> > > Qwen2.5-32B-Instruct|1|24.58|32.29|25.40
> > > -|0|15.77|27.78|11.99
> > > -|-1|3.46|6.75|5.15
> > > -|-2|0.77|6.65|2.19
> > > -|-3|0.61|6.64|2.43
> > >
> > > The suppression patterns are consistent with the factors discussed in Appendix A.1.5. Because multilingual processing involves **distributed, partially redundant neuron groups**, suppressing only a subset of neurons allows the remaining neurons to compensate under mild perturbations. In such settings, stronger (including negative) scale factors are often required to reveal clear behavioral changes. Even under these dynamics, the steady rouge score decline across both models shows that the **targeted neurons make a genuine causal contribution to language production**.
> > > ### W4 & Q3: Quantitative comparisons and human evaluations of neuron description
> > > We recruited five volunteers with master’s degrees in computer science and asked them to independently describe neurons from the level-1 groups of the Arabic, Japanese, and Vietnamese circuits. Volunteers generated descriptions based on each neuron’s top activation samples, with the order of samples shuffled to introduce variability. We then compared these human-generated descriptions with the LLM-generated ones using the Detection method described in Appendix 3.3. The resulting precision scores for both human and LLM descriptions are summarized below.
> > >
> > > Description Generator|Ar|Ja|Vi|Avg
> > > -|-|-|-|-
> > > LLM|0.67|0.73|0.69|0.70
> > > Human|0.66|0.74|0.67|0.69
> > >
> > > An example of human-generated description and LLM-generated description is shown below:
> > > ```
> > > Neuron: 14. 1732
> > > Human 1: Multilingual tokens about geographic or location names
> > > Human 2: Words that denote places, regions, or other geographic locations in different languages
> > > Human 3: Multilingual tokens that talk about places or location names
> > > Human 4: Tokens in different languages that are about geographic spots or location names.
> > > Human 5: Words from various languages that talk about cities, countries, or locations.
> > > LLM: tokens following geographic or location names in multiple languages
> > > ```
> > > We find that the LLM’s performance closely matches human annotators, with an average precision of 0.70, indicating that the generated descriptions are reliable across languages. This alignment suggests that the LLM captures the key features of neuron behavior in a way that is consistent with human judgment. Overall, the results show that interpretive variability is limited and that LLM-based descriptions provide a stable and effective summary of neuron function.
> > >
> > > We additionally assess the stability of the LLM-based circuit descriptions for Arabic by varying both the random seed and the explainer model, and report the corresponding precisions below:
> > >
> > > Random seed|Explainer model|Precision
> > > -|-|-
> > > 0|DeepSeek-v3.2-Exp|0.67
> > > 42|DeepSeek-v3.2-Exp|0.67
> > > 42|Qwen3-max|0.68
> > >
> > > These results show that the precision remains highly consistent across different parameter settings, indicating that the LLM-based descriptions are robust to both seed variation and explainer choice.

---

> > > > ### Author Response · Authors · 2025-11-24
> > > > **Reply to Reviewer NJME Part [04/04]**
> > > >
> > > > ### W5.1 & Q6: Human evaluation on loanwords or named entities
> > > >
> > > > For the LCB benchmark, we filter out samples with lpr=0.0 and manually check the model’s output. The evaluation already uses a common words list to filter out loanwords and named entities, ensuring that most code-switched samples are correctly identified. Some samples were classified as code-switched by the rules but were actually non-code-switched according to human annotators, which we attribute to Unicode-encoded placeholders. However, such cases are rare and do not significantly affect the results.
> > > >
> > > > For our custom benchmark, we use the same common words list to filter out loanwords and named entities and ensure that code-switched fragments do not appear in the question to avoid misclassification. After manual review, we found minimal misjudgments, ensuring the reliability of the LCR metric.
> > > >
> > > > These measures help maintain the robustness and accuracy of our evaluation.
> > > >
> > > > ### W5.2: Code release
> > > >
> > > > We have made our code public on anonymous github and specified in the supplementary material: https://anonymous.4open.science/r/hierarchical_attribution_patching/README.md
> > > >
> > > > ### W5.3 & Q7: Computation cost (GPU-hours)
> > > >
> > > > We list the computational cost for Qwen2.5-7B-Instruct during a complete procedure for our circuit discovery and interpretation framework below.
> > > >
> > > > Step|Approximate Time|Second Per Neuron
> > > > -|-|-
> > > > Level 1 attribution patching|20 s|1.00
> > > > Level 2 attribution patching|2 mins|0.91
> > > > Level 3 attribution patching|8 mins|0.97
> > > > Level 4 attribution patching|10 mins|1.32
> > > > Level 5 attribution patching|17 mins|1.05
> > > > Neuron's top activation samples acquisition (6 languages, 5 level)|1~1.5 hrs|2.56
> > > > Neuron description|27 mins|0.91
> > > > Neuron grouping|53 mins|1.82
> > > > Total|192 mins|10.54
> > > >
> > > >
> > > > ### Q5: Quantitative comparisons with Sparse Autoencoder or ATP* frameworks?
> > > >
> > > > Providing a quantitative comparison with the Sparse Autoencoder framework requires training a sparse autoencoder on all MLPs within the entire model. Unfortunately, this is computationally expensive and beyond our GPU resources. On the other hand, we agree that it is a valuable direction for future work. So we try our best to list a comparison of the three methods below.
> > > >
> > > > | Item | Ours | AtP* | SAE |
> > > > |------|:----:|:----:|:---:|
> > > > | **Description** | Iteratively expands neuron sets across layers; supports hierarchical circuit extraction | Provides refined importance signals for **single-level** component identification | Decomposes activations into interpretable feature directions; **no hierarchical recovery** |
> > > > | **GPU Cost** | **~20 sec** | ~20 sec | High (training required) |
> > > > | **Identify functional neuron groups without auxiliary model training** | ✓ | ✓ | ✗ |
> > > > | **Identify multi-level circuit structure** | ✓ | ✗ | ✗ |
> > > >
> > > >
> > > > Generally speaking, our hierarchical attribution patching method uniquely combines scalability, training-free operation, and the ability to recover multi-level circuit structure, offering practical advantages over both AtP* and SAE within the scope of our study.
> > > >
> > > > > [1] Did Aristotle Use a Laptop? A Question Answering Benchmark with Implicit Reasoning Strategies. Geva et al. TACL 2021.
> > > > >
> > > > > [2] Tracing and Dissecting How LLMs Recall Factual Knowledge for Real World Questions. Wang et al. ACL 2025.
> > > > >
> > > > > [3] How Interpretable are Reasoning Explanations
> > > > from Prompting Large Language Models?. Yeo et al. NAACL 2024.
> > > > >
> > > > > [4] Measuring Mathematical Problem Solving With the MATH Dataset. Hendrycks et al. NeurIPS 2021.
> > > > >
> > > > > [5] Interpreting and Improving Large Language Models in Arithmetic Calculation. Zhang et al. ICML 2024.
> > > > >
> > > > > [6] Towards Understanding Sycophancy in Language Models. Sharma et al. ICLR 2024.
> > > > >
> > > > > [7] From Yes-Men to Truth-Tellers: Addressing Sycophancy in Large Language Models with Pinpoint Tuning. Chen et al. ICML 2024
> > > > >
> > > > > [8] Sparse Feature Circuits: Discovering and Editing Interpretable Causal Graphs in Language Models. Marks et al. ICLR 2025
> > > > >
> > > > > [9] Identifying Query-Relevant Neurons in Large Language Models for Long-Form Texts. Chen et al. AAAI 2024

---

### Official Review · Reviewer_MZtA · 2025-11-01

**Soundness:** 3
**Presentation:** 3
**Contribution:** 3
**Rating:** 6
**Confidence:** 3

**Summary:**

The paper investigates unintended code-switching in multilingual LLMs, a phenomenon where models unexpectedly switch between languages during generation. The authors propose a causal circuit discovery framework that identifies, interprets, and groups multilingual neurons into circuits governing linguistic behavior. Using attribution patching, they reveal two main circuit regimes: (1) language circuits that detect language patterns, and (2) semantic circuits that represent language-agnostic meaning. They argue that code-switching arises from competition between these circuits when semantic activations override language-specific ones. Experiments on Qwen2.5-7B and Llama3.1-8B show that fine-tuning a small set of language-specific neurons (≈0.02%) reduces code-switching by ~20% without hurting task performance.

**Strengths:**

- Novel mechanistic approach: The paper introduces a scalable, neuron-level causal tracing method that extends attribution patching, addressing a gap in multilingual interpretability research.

- Insightful hypothesis: It identifies competition between semantic and linguistic circuits as the mechanistic source of code-switching, a plausible and interpretable explanation.

- Strong empirical results: The selective neuron fine-tuning experiments are efficient and demonstrate measurable improvement (20.8% reduction) in code-switching.

- Method generality: The framework is model-agnostic and tested across two different LLMs and several languages, showing robustness.

- Clear connection to interpretability: The combination of automatic neuron description and grouping makes a step toward more interpretable circuit-level analysis.

**Weaknesses:**

- Overstated causal claims: The method’s causal validity is not convincingly demonstrated—gradient-based attribution and patching only provide correlational evidence under strong linearity assumptions.

- Limited theoretical grounding: The “competition between semantic and language circuits” hypothesis lacks rigorous formalization and is supported mainly by qualitative visualizations.

- Reproducibility concerns: Although implementation details are mentioned, many core components (e.g., neuron labeling prompts, clustering procedure, dataset construction) depend on opaque LLM judgments.

**Questions:**

as suggestion you might also want to use a token-level or sequence-level code-switching detection approaches.
For instance, for code-switch language identification: https://aclanthology.org/2024.acl-short.43/ which uses models like https://aclanthology.org/2023.findings-emnlp.410/ to automatically detect and classify code-switched content.

---

> ### Author Response · Authors · 2025-11-24
> **Reply to Review MZtA Part [01/03]**
>
> ### W1: Experimental evidence for causal validity.
> **We observe a consistently high first-level overlap**, typically 12–18 out of 20 neurons—between attribution patching and activation patching across models and languages. Attribution patching estimates the causal effect of a neuron’s contribution on the output under a linearized neighborhood of the activation manifold.
>
> In order to test whether attribution patching identifies neurons that genuinely exert causal influence, we computed the overlap between the **top-20 neurons** selected by attribution patching and activation patching. Because full hierarchical activation patching is extremely time-consuming (>2 days per level), we evaluate the first-level overlap only.
>
> Across two models and three language circuits, we observe consistent, above-chance overlap:
> Model|Vietnamese|Arabic|Japanese
> -|-|-|-
> Qwen2.5-7B-Instruct|80%|60%|65%
> LLaMA3.1-8B-Instruct|75%|90%|70%
>
> The substantial overlaps we observe show that the neurons selected by attribution interventions strongly align with the neurons validated as causal via activation-level interventions. This is also supported by the prior work using gradient-based interventions for circuit discovery ([1][2]).
> ### W2: Theoretical grounding
> Thanks for the constructive advice. Below we privide a preliminary theoretical grounding of the ''competition between semantic and language circuits''.
>
> Let $i$ represent the index of a Chinese word in the vocabulary, and $j$ the index of the corresponding French word (e.g., at position $i$, we have "非洲" and at position $j$, "Afrika").
>
> Define $e_i$ as the one-hot vector in $\mathbb{R}^V$ where the $i$-th position is 1 and the rest are 0, and $V$ is the size of the vocabulary. The logits corresponding to the Chinese and French tokens can then be written as:
> $$\text{Logits}[\text{非洲}] = \text{Logits}^T \cdot e_i, \quad \text{Logits}[\text{Afrika}] = \text{Logits}^T \cdot e_j.$$
>
> Given a neuron with activation $a$, let $a_0 = a_{\text{clean}}$ represents the activation for a Chinese input, and $a_1 = a_{\text{corrupt}}$ represents the activation for a French input. Let $x_0$ denote the first $n-1$ tokens of the Chinese input and $x_1$ denote the first $n-1$ tokens of the French input.
>
> $\text{Logits}(a, x)$ represents the logits produced by the model when the input is $x$ and the targeted neuron activation is forced to be $a$. So:
> - $\text{Logits}_{\text{clean}} = \text{Logits}(a_0, x_0)$ represents the model’s original prediction on the clean Chinese input, using the neuron activation naturally induced by the Chinese context.
> - $\text{Logits}_{\text{corrupt}} = \text{Logits}(a_1, x_1)$ represents the model’s prediction on the fully corrupted French input, using the neuron activation naturally induced by the French context.
> - $\text{Logits}_{\text{patch}} = \text{Logits}(a_0, x_1)$ represents a causal intervention where the French input is used but the neuron activation is overwritten with the activation from the Chinese instance, thereby isolating the causal effect of that neuron on the final logits.
>
> Thus, the logit differences (LD) are:
>
> $$
> LD_{\text{patch}} = Logits_{patch}[非洲] - Logits_{patch}[Afrika]
> $$
>
> $$
> = \text{Logits}_{\text{patch}}^T \cdot (e_i - e_j) = \text{Logits}(a_0, x_1)^T \cdot (e_i - e_j) ,
> $$
>
> $$
> LD_{\text{corrupt}} = \text{Logits}(a_1, x_1)^T \cdot (e_i - e_j) ,
> $$
>
> $$
> LD_{\text{clean}} = \text{Logits}(a_0, x_0)^T \cdot (e_i - e_j).
> $$
>
> Given the formula above, the metric $m$ in Equation (2) can be written as:
>
> $$
> m = \frac{(\text{Logits}(a_0, x_1) - \text{Logits}(a_1, x_1))^T \cdot (e_i - e_j)}{(\text{Logits}(a_0, x_0) - \text{Logits}(a_1, x_1))^T \cdot (e_i - e_j)}.
> $$
>
> By treating $LD_{\text{corrupt}}$ and $LD_{\text{clean}}$ as constants when differentiating, we obtain:
>
> $$
> \frac{\partial m}{\partial a} = \frac{1}{(Logits(a_0, x_0) - Logits(a_1, x_1))^T \cdot (e_i - e_j)} \left( \frac{\partial Logits(a, x)}{\partial a} |_{a = a_0, x = x_1} \right)^T \cdot (e_i - e_j).
> $$
>
> Hence, the attribution $\text{Attr}$ is:
>
> $$
> \text{Attr} = \frac{a_0 - a_1}{(\text{Logits}(a_0, x_0) - \text{Logits}(a_1, x_1))^T \cdot (e_i - e_j)} \left( \frac{\partial \text{Logits}(a, x)}{\partial a} \bigg|_{a = a_0, x = x_1} \right)^T \cdot (e_i - e_j).
> $$
>
> The activation $a$ represents the **neural response to a specific input**, which influences the output logits. Specifically, $\frac{\partial m}{\partial a}$ represents **how much a change in the activation $a$ affects the difference between the logits for the Chinese and French tokens**. This term measures the contribution of a particular neuron to the output, capturing the extent to which the neuron’s activation influences the logit difference, or the “decision” for the model’s output.
>
> Now, let’s introduce two types of neurons: one representing **language-specific patterns** whose activations are denoted by $a_l$, and the other representing **semantic patterns** whose activations are denoted by $a_s$.

---

> > ### Author Response · Authors · 2025-11-24
> > **Reply to Review MZtA Part [02/03]**
> >
> > Notably, the denominator $(\text{Logits}(a_0, x_0) - \text{Logits}(a_1, x_1))^T \cdot (e_i - e_j)$ is independent of which neuron we choose for patching (because it doesn't require patching and is calculated over fixed input), so it cancels out when comparing the ratio. Thus, the ratio of attributions for the language and semantic neurons is:
> >
> > $$
> > Attr_l = (a_{l,0} - a_{l,1}) (\frac{\partial Logits(a_l, x)}{\partial a_l}|_{a=a_l,0,x=x_1})^\top\cdot(e_i-e_j).
> > $$
> >
> > $$
> > Attr_s = (a_{s,0} - a_{s,1}) (\frac{\partial Logits(a_s, x)}{\partial a_s}|_{a=a_s,0,x=x_1})^\top\cdot(e_i-e_j).
> > $$
> >
> > To simplify, we can apply a first-order Taylor expansion around the numerator and denominator. For the language neuron and semantic neuron, we approximate:
> > $$
> > \frac{Attr_l}{Attr_s} \approx \frac{(Logits(a_{l,0},x_1)-Logits(a_{l,1},x_1))^T \cdot (e_i - e_j)}{(Logits(a_{s,0},x_0)-Logits(a_{s,1},x_1))^T \cdot (e_i - e_j)}
> > $$
> >
> > $$
> > = \frac{(Logits_{\text{patch-language}}[\text{非洲}] - Logits_{\text{corrupt}}[\text{非洲}]) - (Logits_{\text{patch-language}}[\text{Afrika}] - Logits_{\text{corrupt}}[\text{Afrika}])}{(Logits_{\text{patch-semantic}}[\text{非洲}] - Logits_{\text{corrupt}}[\text{非洲}]) - (Logits_{\text{patch-semantic}}[\text{Afrika}] - Logits_{\text{corrupt}}[\text{Afrika}])}.
> > $$
> >
> > In **Non-Code-Switched Case:**
> > * **Numerator**: When patching the **language neuron**, the **Chinese token** “非洲” exhibits a significant **increase** in probability, while the **French token** "Afrika" exhibits a significant **decrease**, resulting in a large positive numerator.
> > * **Denominator**: After patching the **semantic neuron**, the probability for the **Chinese token** “非洲” should experience a slight increase, while the probability for the **French token** "Afrika" should experience a slight decrease. As a result, the first term in the denominator is a small positive value, and the second term is a small negative value, leading to a smaller positive denominator and a large attribution ratio.
> >
> > In **Code-Switched Case:**
> > * **Numerator**: In the code-switched scenario, patching the **language neuron** still causes an **increase** for “非洲”, but the **decrease** for "Afrika" is less pronounced. Consequently, the numerator is smaller compared to the normal case.
> > * **Denominator**: When patching the **semantic neuron**, the changes in probability for both tokens are minimal, resulting in a smaller attribution for the **semantic neuron** relative to the **language neuron**. As a result, the overall attribution ratio approaches 1.
> >
> > The experimental results in Section 5.3 align with this interpretation. In Table 2, the attribution ratio for non-code-switched samples is 2.37, dominated by language circuits, while for code-switched samples, the ratio decreases to 1.21, indicating a shift toward semantic circuits. Furthermore, as shown in Table 3, intervening on semantic circuits significantly reduced the probability of the target code-switched token, demonstrating that semantic circuits play a more prominent role in code-switching.
> >
> > This experimental outcome supports the idea that code-switching arises from the weakening of language circuit dominance and the increased influence of semantic circuits. These results directly validate our theoretical hypothesis that semantic circuits compete with language circuits, with semantic circuits becoming more dominant in code-switched contexts.
> >
> > Thanks for your insightful feedback. We have added this  discussion in our revised manuscript.
> >
> > ### W3: Reproducibility of prompts and dataset construction.
> > We thank the reviewer for raising concerns regarding reproducibility of components such as neuron labeling prompts, the clustering procedure, and dataset construction.
> > All three components are fully specified in the paper:
> >  - Neuron labeling prompts are provided in Appendix A.4.1, and we adopt the default sampling params of the explainer model DeepSeek-V3.2-Exp, with temperature=0.0.
> >  - The clustering procedure  is described in Section 4.3, with grouping prompt in Appendix A.4.2.
> >  - Dataset construction details are documented in Appendix A.2, with no LLM usage introduced.
> >
> > To ensure reproducibility, we conducted additional experiments that evaluate the stability of both the LLM-generated neuron descriptions and clustering.
> >
> > For neuron descriptions, we recruited five volunteers with master’s degrees in computer science and explained the neuron-description procedure in detail. We then asked them to independently label neurons from a specific level within  different language circuits. Concretely, we selected the level-1 neurons of the Arabic, Japanese and Vietnamese circuit and asked the volunteers to produce descriptions using the neurons’ top activation samples. To simulate different reproduction scenarios, we also randomly shuffled the order of each neuron’s top activation samples during the labeling process.

---

> > > ### Author Response · Authors · 2025-11-24
> > > **Reply to Review MZtA Part [03/03]**
> > >
> > > After collecting the human-written descriptions, we computed the precision of both the human descriptions and the LLM-generated descriptions using the Detection method. Results see below:
> > >
> > > Description Generator|Ar|Ja|Vi|Avg
> > > -|-|-|-|-
> > > LLM|0.67|0.73|0.69|0.70
> > > Human|0.66|0.74|0.67|0.69
> > >
> > > An example of human-generated descriptions and LLM-generated description is shown below:
> > > ```
> > > Neuron: 14. 1732
> > > Human 1: Multilingual tokens about geographic or location names
> > > Human 2: Words that denote places, regions, or other geographic locations in different languages
> > > Human 3: Multilingual tokens that talk about places or location names
> > > Human 4: Tokens in different languages that are about geographic spots or location names.
> > > Human 5: Words from various languages that talk about cities, countries, or locations.
> > > LLM: tokens following geographic or location names in multiple languages
> > > ```
> > > Overall, the LLM’s performance closely matches that of human annotators, indicating that the generated neuron descriptions are both stable and reliable across languages.
> > >
> > > We further assessed the reliability of the **clustering procedure** by examining the level-1 groups in the Arabic, Japanese, and Vietnamese circuits. For each group, we randomly sampled up to five neurons and provided their descriptions, together with the full list of level-1 cluster names, to five volunteers with a computer science background. Volunteers were asked to decide which cluster each neuron most likely belonged to, or to select “none of the above” if no cluster appeared suitable. We then compared these human-assigned labels with the actual cluster assignments to evaluate how consistently the clustering reflects human-interpretable structure. The resulting consistency scores are shown in the table below.
> > >
> > > Language|Precision|
> > > -|-
> > > Ar|16/20|
> > > Ja|13/16
> > > Vi|13/17
> > >
> > > The results demonstrate that the clustering procedure yields high agreement between the human-assigned and true cluster labels, confirming the reliability of our clustering method. This further validates the robustness and reproducibility of the neuron labeling and clustering components of our framework.
> > > ### Q1: Additional methods for code-switch language identification
> > > Thank you for the suggestion regarding alternative code-switching detection. We added an evaluation using the GlotLID model from [3]. Because GlotLID processes only newline-free text, we split each completion on `\n` and applied the model to each segment. Following [3], we assign a language label only when its predicted probability exceeds 0.9, and classify a completion as non–code-switched when all segments share the same high-confidence label. We report the resulting GlotLID-based language consistency rate (LCR) alongside our original LPR metric on LCB in the table below.
> > >
> > > Model|Metric|Ar|Es|Fr|Ja|Ru|Zh|Vi|Id|Avg
> > > -|-|-|-|-|-|-|-|-|-|-
> > > Qwen2.5-7B-Instruct|LPR|96.41|96.67|95.18|88.78|98.08|92.10|97.38|82.96|93.45
> > > -|LCR|72.43|87.03|82.37|69.00|90.90|91.80|95.00|90.30|84.85
> > > LLaMA3.1-8B-Instruct|LPR|99.25|97.63|97.86|96.00|98.40|94.55|99.70|83.50|95.99
> > > -|LCR|63.60|87.77|92.20|80.60|94.30|86.80|94.30|90.70|86.28
> > >
> > > While our overall results show strong agreement between the two metrics, we do observe noticeable gaps for certain languages, and we investigated these cases carefully. Our analysis reveals that GlotLID often provides more fine-grained distinctions than what our Unicode-based method is designed to capture.
> > >
> > > For example, an Arabic completion may be labeled simultaneously as `ary_Arab` (Moroccan Arabic) and `arb_Arab` (Modern Standard Arabic). These languages differ only in a small number of script-specific characters, but at the Unicode level they are all simply Arabic script, which naturally creates discrepancies between the two methods. Therefore, GlotLID’s finer-grained distinctions naturally lead to different labels even when the underlying script is effectively the same.
> > >
> > > To further address this, we cluster language labels that correspond to different variants or dialects of the same underlying language and recompute the LCR. The results are shown below.
> > >
> > > Model|Metric|Ar|Es|Fr|Ja|Ru|Zh|Vi|Id|Avg
> > > -|-|-|-|-|-|-|-|-|-|-
> > > Qwen2.5-7B-Instruct|LPR|96.41|96.67|95.18|88.78|98.08|92.10|97.38|82.96|93.45
> > > -|LCR|95.30|94.77|95.00|84.50|95.50|90.05|94.20|81.60|91.37
> > > LLaMA3.1-8B-Instruct|LPR|99.25|97.63|97.86|96.00|98.40|94.55|99.70|83.50|95.99
> > > -|LCR|99.53|96.23|94.23|88.30|99.20|91.70|97.30|92.70|94.90
> > >
> > > After consolidating these variants, the recomputed LCR closely matches the LPR reported in our paper, confirming that the remaining discrepancies arise primarily from GlotLID’s finer-grained distinctions rather than inconsistencies in our evaluation.
> > >
> > > >[1] Sparse Feature Circuits: Discovering and Editing Interpretable Causal Graphs in Language Models. Marks et al.
> > > >
> > > >[2] Identifying Query-Relevant Neurons in Large Language Models for Long-Form Texts. Chen et al.
> > > >
> > > >[3] GlotLID: Language Identification for Low-Resource Languages. Kargaran et al.

---

> ### Comment · Reviewer_MZtA · 2025-11-28
>
> Thank you for your reply. I will keep my positive assessment. I would have increased it further, but it seems I can’t revise my assessment anymore.
>
> > Because GlotLID processes only newline-free text, we split each completion on \n and applied the model to each segment.
>
> For the language ID model, you can replace “\n” with a space.

---

### Author Response · Authors · 2025-11-30
**Summary Comment for ACs**

Dear Reviewers, Area Chairs, and Program Chairs,

We sincerely thank all four reviewers for their constructive comments and insightful questions, which helped us refine our work.

**Acknowledgement of Review Strengths**

**[Problem Importance]**

- **Reviewer EbfB**: This study conducts analysis on the cause of language-switching; The research theme is important for safety and intriguing.

**[Method Novelty]**

- **Reviewer MZtA**: Novel mechanistic approach: The paper introduces a scalable, neuron-level causal tracing method that extends attribution patching, addressing a gap in multilingual interpretability research.
- **Reviewer NJME**: Novel application of mechanistic interpretability. Code-switching is an interesting and underexplored multilingual behavior, and analyzing it through causal circuit tracing is conceptually fresh.

**[Promising Results]**

- **Reviewer MZtA**: Strong empirical results: The selective neuron fine-tuning experiments are efficient and demonstrate measurable improvement (20.8% reduction) in code-switching.
- **Reviewer inWb**: Overall, the paper balances mechanistic insight with empirical robustness, offering one of the **most convincing analyses to date** of language competition inside multilingual LLMs.
- **Reviewer EbfB**: The authors conducted a comprehensive experiments using two models, multiple languages, multiple benchmarks, ensuring the findings and effectiveness.

**[Clear Presentation]**
- **Reviewer NJME**:
  - Detailed technical description. The paper provides pseudo-code (Algorithm 1), datasets, and hyperparameters, making the procedure relatively transparent.
  - Clear organization. The presentation of results—particularly Figures 1–3 and Tables 1–5—is systematic and accessible.
- **Reviewer inWb**: The paper is also methodologically elegant and transparent.

During the discussion period, we made every effort to provide clear and detailed responses to all reviewer comments, including supplementary experiments where feasible. Below, we briefly summarize our clarifications (with full details included in our rebuttals):

**[Mathematical Derivation]**: We provided full derivations of the competition dynamics between the language sub-circuit and semantic sub-circuit, offering mathematical support for our explanation of code-switching. We also included a complete derivation of the hierarchical attribution patching process in the revised manuscript.

**[Causal Evidence]**: We performed comparisons between hierarchical attribution patching and standard activation patching. The results show close alignment between the two, with both identifying similar key neurons, offering strong causal evidence for the effectiveness of our method.

**[Method Generality]**: We demonstrated that our approach generalizes across model scales and architectures (Mistral-7B-Instruct-v0.3, Gemma2-9B-Instruct, Qwen2.5-32B-Instruct) and across domains (reasoning, arithmetic, sycophancy), confirming the robustness and broad applicability of our framework.

Due to the special circumstances this year, extended discussion was limited. However, we did receive encouraging follow-up from Reviewer MZtA, who expressed an intention to raise their score from 6 to 8 after reviewing our rebuttal and additional analyses.

We are deeply grateful for the reviewers’ constructive and valuable recommendations. We will incorporate all suggested clarifications and additional experiments into the final version of the paper. We sincerely hope that our responses adequately address the reviewers’ concerns.

Thank you very much for your time and consideration.

Best regards,

Authors of #11933

---

### Meta-Review · Area_Chair_BvWh · 2026-01-04

**Summary:**

This work presents a mechanistic interpretability framework to uncover the neural causes of unintended code-switching in multilingual LLMs by identifying and analyzing competing language-specific and semantic circuits at the neuron level. The authors show that code-switching emerges from competition between these circuits and can be substantially reduced by fine-tuning a very small subset of neurons. Experiments on Qwen2.5-7B and Llama3.1-8B show that fine-tuning a small set of language-specific neurons (≈0.02%) can reduce code-switching by ~20%.

Although this study is quite interesting, it relies on relatively strong assumptions, and its theoretical component needs to be further strengthened. Some of the key conclusions require more solid experimental and theoretical support.

**Reviewer Concerns:**

The rebuttal has addressed the  Reviewer MZtA's concern.

**Reviewer Scores:**

Reviewer MZtA stated that they will raise the score.

---

### Decision · Program_Chairs · 2026-01-26

Reject